# SCALING LAWS AND SYMMETRY, EVIDENCE FROM NEURAL FORCE FIELDS

**Khang Ngo** [1] [2]**, Siamak Ravanbakhsh** [1] [2]
[1] Mila - Quebec AI Institute, [2] School of Computer Science, McGill University
{khang.ngo, siamak.ravanbakhsh}@mila.quebec

## ABSTRACT

We present an empirical study in the geometric task of learning interatomic potentials, which shows equivariance matters even more at larger scales; we show a clear power-law scaling behaviour with respect to data, parameters and compute with "architecture-dependent exponents". In particular, we observe that equivariant architectures, which leverage task symmetry, scale better than non-equivariant models. Moreover, among equivariant architectures, higher-order representations translate to better scaling exponents. Our analysis also suggests that for compute-optimal training, the data and model sizes should scale in tandem regardless of the architecture. At a high level, these results suggest that, contrary to common belief, we should not leave it to the model to discover fundamental inductive biases such as symmetry, especially as we scale, because they change the inherent difficulty of the task and its scaling laws. [1]

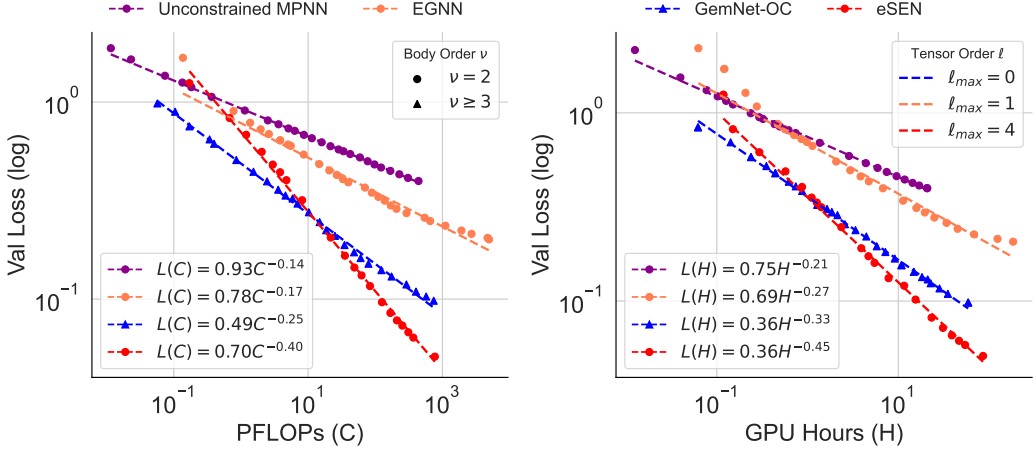

Figure 1: Performance of neural network interatomic potentials follows a power law (linear in log-log space) in training compute (PFLOPs, GPU-hours). The scaling behaviour varies with architectural complexity: the slope of the performance curve improves as the architecture changes from unconstrained to low-order to high-order, implying that performance gaps widen with increasing compute. *Body order $\nu$*: number of nodes whose states define a message within a layer. *Tensor order $\ell$*: order of geometric features processed by the models. **Left**: Empirical scaling laws along the FLOPs-optimal frontier. **Right**: Empirical scaling laws along the train-time-optimal frontier.

## 1 INTRODUCTION

Recent years have witnessed extensive study of neural scaling laws across various machine learning domains, including natural language and vision. The general observation supported by the theory is that test errors exhibit a power-law relationship with the scale of training data points, model

---

[1]Code is available at https://github.com/nnkhang19/scaling-laws-and-symmetry

parameters, and the amount of compute in floating-point operations (FLOPs). These laws identify the optimal scaling of model size with dataset size for a given compute budget, enabling an optimal use of resources at scale.

A common view is that the scaling behaviour is consistent across various expressive architectures for a task—i.e., the choice of architecture can only increase or decrease the loss by a multiplicative factor that remains constant across sufficiently large scales. This belief is supported by both theoretical results (Sharma & Kaplan, 2022; Bahri et al., 2024) and empirical studies in different domains (Ahmad & Tesauro, 1988; Hestness et al., 2017), including language (Kaplan et al., 2020; Hoffmann et al., 2022) and vision (Zhai et al., 2022), and it is further reinforced by Sutton's bitter lesson (Sutton, 2019), which highlights that attempts to encode inductive biases, such as symmetry, explicitly are often outperformed in the long run, since models can learn these structures on their own when scaled.

The specific inductive bias of symmetry, and in particular Euclidean and rotational symmetry have been successfully leveraged in many domains, including for molecular force fields. The success of these networks is often attributed to their improved generalization and robustness to out-of-distribution data (Batatia et al., 2022; Petrache & Trivedi, 2023). However, one may argue that equivariant networks are harder to scale as their specialized operations, such as tensor products, spherical harmonics (Thomas et al., 2018; Anderson et al., 2019; Liao & Smidt, 2023), or high-order message passing (Gasteiger et al., 2020b; Klicpera et al., 2021), are complex and computationally expensive. At the same time, several works in protein folding (Abramson et al., 2024), molecule conformer generation (Wang et al., 2024), and Neural Network Interatomic Potentials (NNIPs) (Deng et al., 2023; Qu & Krishnapriyan, 2024; Rhodes et al., 2025) demonstrate that non-equivariant networks can perform well in geometric tasks. Brehmer et al. (2025) also show that non-equivariant architectures trained with data augmentation can perform on par with their equivariant counterparts when given sufficient compute. All of this paints a picture in favour of forgoing equivariance and scaling simpler non-equivariant models.

This paper presents a careful empirical study that questions this growing mindset and shows that equivariance matters even more as we scale. We report a clear architecture-dependent scaling exponent in model size, data size, and compute, for several widely used scalable NNIPs architectures that encode rotational and permutation symmetry to varying degrees. This translates to a performance gap that grows with scale, favouring scalable models with a higher-order symmetry bias at larger scales; see fig. 1.

Our target domain for this study has witnessed a growing number of deep learning techniques for predicting quantum properties of atomistic systems in recent years, where neural models approximate computationally demanding ab initio calculations, such as density functional theory. The most promising progress is being made on NNIPs, which map molecular systems to their energies and forces. NNIPs' foundation models are unlocking new possibilities through efficient and accurate molecular dynamics, and our findings in this domain identify the most promising direction for the design of models that are trained at scale.

## 1.1 KEY FINDINGS AND CONTRIBUTIONS

In this work, we conduct comprehensive scaling-law experiments, drawing on best practices and insights from prior work on the expressive power of (geometric) message passing neural networks (MPNNs) (Loukas, 2020; Joshi et al., 2023), maximal update parametrization ($\mu$P) (Yang et al., 2021), and compute-optimal scaling (Hoffmann et al., 2022). Our key findings are:

- **Clear power law scaling.** Message-passing NNIPs obey power-law scaling with respect to compute, data, and model size. For compute, unlike prior studies that report only FLOPs within architectures, we characterize scaling with both FLOPs and wall-clock training time (GPU-hours). Given that equivariant networks can be less GPU-friendly, this approach provides a more complete view for practical purposes. While prior work in geometric domains has shown architecture-dependent scaling with respect to only the dataset size (Batzner et al., 2022), to our knowledge, none of them provide a complete and comparable picture to ours.

- **Architecture-dependent exponents.** Power-law exponents increase as the "degree" of equivariance grows, from non-equivariant (unconstrained) models to lower- to higher-order equivariant designs.

- *Compute-optimal scaling.* We find that the power-law exponents for dataset and model size in a compute-optimal scaling are similar across non-equivariant and equivariant architectures of different representation degrees. This means that a compute-optimal scaling should increase the model and dataset size in tandem; this mirrors the findings of Hoffmann et al. (2022) in natural language.

- *Multi-epoch training and data-augmentation.* While our main results consider a single epoch regime, we show that the same scaling laws hold across tens of epochs in a multi-epoch setting. This is because, at scale – even with 1% of our training set – the effect of overfitting is negligible. For non-equivariant models, data augmentation is required to avoid overfitting and maintain the scaling coefficients. We also consider inference-time augmentation for the unconstrained model, and show that it only changes the multiplicative coefficient (rather than the exponent) in the scaling law, and its benefit saturates quickly with the number of augmentations for this task.

- *Scaling effect of symmetry loss.* Enforcing symmetry through loss does NOT seem to provide the same benefits as having an equivariant architecture.

- *Trend in optimal depth.* For a fixed parameter and compute budget, the optimal depth of the network is correlated with the "degree" of equivariance among the architectures we studied; with equivariant networks, the benefit of depth saturates at higher values, and for higher rotation order networks this value grows larger; this corroborates the claims of Joshi et al. (2023); Jia et al. (2020).

**Organization.** The rest of the paper is organized as follows. Section 2 outlines problem setup and symmetry constraints through architectures and loss. Section 3 discusses our experiments, including key hyperparameters and scaling strategies. Section 4 presents our results and their analysis. Section 5 concludes with an emphasis on limitations of our work, and important directions for near-future work. Details on related works, background, experimental setup, and additional results are included in the Appendix.

## 2 SETUP

An atomistic system can be represented as a point cloud $X = \{(z_1, x_1), \ldots, (z_n, x_n)\}$, where $z_i \in \mathbb{N}$ and $x_i \in \mathbb{R}^3$ are the the atomic number and the position respectively. The potential energy $e(X)$ is a scalar that is invariant to global translations and rotations, while forces $f_i(X) = -\partial e(X)/\partial x_i$ are vectors that are translation-invariant and rotation-equivariant. The task of our NNIPs is to train a neural network $\phi_\theta : \mathbb{N} \times \mathbb{R}^{3 \times n} \to \mathbb{R}^{1+3 \times n}$ that takes $X$ as input and predicts the potential energy (scalar) and atom-level forces, one for each atom – i.e., $\phi_\theta : X \mapsto (e_\theta(X), \{f_{\theta,1}(X), \ldots f_{\theta,n}(X)\})$. While it is sufficient to learn the energy for predicting conservative forces, direct force prediction is significantly more scalable. Using this approach enables one to benefit from the dense signal during pre-training. In post-training, the force prediction can be removed and the model can be fine-tuned to predict conservative forces through backpropagation via the predicted energies, ensuring a good balance between computational cost and accuracy (Bigi et al., 2025; Fu et al., 2025). We minimize the per-atom mean absolute error (MAE) and mean squared error of forces (Fu et al., 2025; Wood et al., 2025):

$$\mathcal{L}(\phi_\theta, X) = \frac{\lambda_e}{n} \left\| e_\theta(X) - e(X) \right\|_1 + \frac{\lambda_f}{n} \sum_{i=1}^{n} \left\| f_{\theta,i}(X) - f_i(X) \right\|_2, \tag{1}$$

with $\lambda_e, \lambda_f > 0$ are the coefficients that control the relative importance of energy and force predictions; we use $\lambda_e = \lambda_f$.

### 2.1 ARCHITECTURES

Since we observed instability issues when scaling vanilla transformers for this task, we focused on message-passing architectures. Here, in addition to a basic unconstrained MPNN, following the classification of MPNNs in Joshi et al. (2023), we considered three widely adopted equivariant architectures that cover various body and tensor orders. The body order corresponds to $S_n$ representations, and refers to the number of nodes participating in a message function. The tensor order $\ell$ corresponding to $SO(3)$ representations, and denotes the order of the geometric tensor embeddings processed by each model. Below we briefly enumerate these; for more background on these architectures, see appendix C:

1. ***unconstrained***: a vanilla MPNN that directly processes geometric features, i.e., relative position vectors, without any symmetry constraints.

2. ***invariant scalars***: geometric message passing neural network (GemNet-OC) (Gasteiger et al., 2022) is a variation of GemNet (Klicpera et al., 2021) adapted for large and diverse molecular dataset. Although it uses invariants such as interatomic distances and angles, and therefore has a tensor order $\ell = 0$, it can approximate equivariant functions from edge-based invariant features, because it performs geometric message passing with two-hop and edge-directional embeddings; see Theorem 3 in Klicpera et al. (2021). Because GemNet-OC incorporates dihedral-angle information, its two-hop messages depend simultaneously on the states of four nodes, and thus it is classified as four-body.

3. ***Cartesian vectors***: E(n)-equivariant graph neural network (EGNN) (Satorras et al., 2021); in particular, the extension of Levy et al. (2023), which allows for more than one equivariant vector channel. We use a specific $\mu$P informed scaling, in which scalar channels scale quadratically wrt number of vector channels, see appendix C for details.

4. ***high-order spherical tensors***: equivariant Smooth Energy Network (eSEN) (Fu et al., 2025), which uses higher-order irreducible representaions of rotation group ($\ell \geq 2$). Unlike other architectures in the same category (e.g., Thomas et al., 2018; Batzner et al., 2022; Liao & Smidt, 2023), we found eSEN more scalable because it uses frame alignment to sparsify the tensor product, allowing it to eliminate Clebsch–Gordan coefficients and to directly parameterize kernels with linear layers (Passaro & Zitnick, 2023).

## 2.2 SYMMETRY LOSS

Symmetry-based losses have been used in different settings from self-supervised learning (Dangovski et al., 2021; Bai et al., 2025), to physics-informed settings (Akhound-Sadegh et al., 2023; Yang et al., 2024), generative modelling (Tong et al., 2025) and symmetry discovery (Escriche & Jegelka, 2025). A canonical choice is a loss that penalizes deviations from equivariance constraints for randomly sampled global transformations (e.g., Kim et al., 2023a; Elhag et al., 2025; Bai et al., 2025):

$$\mathcal{L}_{\text{sym}}(\phi_\theta; x, y) \;=\; \frac{1}{M} \sum_{i=1}^{M} \mathcal{L}\big(\phi_\theta(\rho_{\text{in}}(g_i)\, x),\; \rho_{\text{out}}(g_i)\, y\big), \tag{2}$$

where $g_i \sim \mu_G$ is a sampled from the Haar measure, $\rho_{\text{in}}$ and $\rho_{\text{out}}$ define linear actions on inputs $x$ and targets $y$, respectively, and $\mathcal{L}$ is the task loss. The symmetry-augmented term is added to the base objective in eq. (1) when training $f_\theta$.

For our task, the translation part of the special Euclidean group $SE(3) = SO(3) \ltimes T(3)$ is accounted for by centring the molecule at its center of mass, and the loss is only for the rotation group. [2]

## 3 EXPERIMENTS

### 3.1 DATASET

We conduct our experiments on the OpenMol neutral-molecule subset (Levine et al., 2025), with 34M training samples and 27K held-out validation samples.[3] Treating atom nodes as tokens, the training set corresponds to approximately $D \approx 9.2 \times 10^8$ tokens. Following scaling studies in LLMs, we consider a single-epoch training regime, where each sample is observed exactly once. While a multi-epoch setting can be more practical for AI4Science due to smaller datasets compared to language, our goal was to stay faithful to existing methodologies and avoid possible confounding effects (Muennighoff et al., 2023).

### 3.2 OPTIMIZATION

Following Choshen et al. (2025), which shows that estimating scaling laws from intermediate checkpoints yields more robust results, we track validation losses throughout training and fit these

---

[2]We also tried using regularization that measures invariance by differentiating along infinitesimal generators, similar to Rhodes et al. (2025), but we could not achieve stable training.

[3]We use the neutral subset rather than the full 100M-molecule corpus due to main-memory constraints.

points—excluding the first $1\% - 10\%$ of steps—to a standard scaling-law functional form (Kaplan et al., 2020; Hoffmann et al., 2022). A well-known caveat in scaling-law analyses is the sensitivity to learning-rate schedules, particularly when predefined decay steps are used (Hoffmann et al., 2022; Hu et al., 2024). To address this, we adopt scheduler-free AdamW-style optimizers (Defazio et al., 2024), which not only remove the need for tuning decay schedules but also allow us to capture model training dynamics within a single run—without retraining from scratch at each data ratio or relying on checkpoint restarts (Hu et al., 2024). Crucially, this approach enables more accurate measurement of training time by mitigating hardware-related artifacts and helps us derive scaling laws directly with respect to training time, measured in GPU-hours.

### 3.3 Hyper-parameter Tuning

Investigating scaling behaviours of neural networks necessitates evaluations under optimal conditions for both efficiency and performance. We, therefore, perform systematic experiments to determine critical hyperparameters affecting the scaling behaviours of those architectures. Our analysis include non-equivariant MPNN, EGNN, and GemNet-OC. Due to the higher computational cost of eSEN, we adopt the optimal hyperparameters from Passaro & Zitnick (2023); Fu et al. (2025).

**Learning Rates and Batch Sizes.** When fine-tuning the model, we swept over 12 configurations for each architecture by testing three learning rates, $\{1e-4, 5e-4, 1e-3\}$, and four batch sizes, $\{64, 128, 256, 512\}$. We performed the tuning with approximately one million parameters. As shown in fig. 10, smaller batch sizes and larger learning rates resulted in lower validation losses. This finding about small batch sizes aligns with observations in (Gasteiger et al., 2020a; Frey et al., 2022).

**Saturation Depth.** Depth $d$ and width $w$ (embedding dimension) govern the parameter count $N$. While Kaplan et al. (2020) found Transformer performance depends mainly on $N$ and is independent of architectural factors such as $d$ and $w$, recent MPNNs work shows that depth choice can undermine power-law scaling behaviour (Liu et al., 2024; Sypetkowski et al., 2024). We therefore probe the depth-saturation point on 3D geometric graphs—the depth beyond which validation error no longer improves for fixed capacity (Loukas, 2020). To isolate depth/width from model size, we fix $N \approx 10^6$ and sweep $(d, w)$. As shown in fig. 10, non-equivariant MPNNs degrade with increasing depth (over-smoothing/over-squashing (Topping et al., 2022)), whereas equivariant models continue to improve, with validation losses plateauing at depths $L \in \{12, \ldots, 16\}$, consistent with prior reports (Joshi et al., 2023; Passaro & Zitnick, 2023; Pengmei et al., 2025).

**Infinite-Width Scaling.** Our depth-saturation experiments are motivated by the universality condition for message-passing-based architectures, which requires "sufficient depth" and unbounded width; see Corollary 3.1 in Loukas (2020). For each architecture type, we train a series of models with an increasing number of channels (width) along a scaling ladder. We fix the optimal hyperparameters—including depth, learning rate $\eta^* = 1e{-}3$, and batch size—for $\approx$1M-parameter models with base width $w_{\text{base}}$, and use $\mu$P (Yang et al., 2021) to transfer $\eta^*$ to other widths $w$ via $\eta(w) = \eta^* \cdot \frac{w_{\text{base}}}{w}$. We increase model size until the memory of a single NVIDIA A100 (40 GB) GPU is exhausted. In other words, we keep depth and batch size constant across model sizes and scale the width as high as our hardware allows.

## 4 Scaling Laws

### 4.1 Scaling Compute

Nominal FLOPs are hardware-agnostic, yet equivariant models often have lower GPU utilization, so FLOPs may understate practical cost. We therefore fit scaling laws in both theoretical FLOPs $C$ and wall-clock training hours $H$, training all models on identical hardware; see Appendix D.

**Counting FLOPs.** Following Kaplan et al. (2020); Hoffmann et al. (2022), we define the compute as theoretical FLOPs counting $C$ incurred from training a model of $N$ parameters on $D$ training tokens:

$$C \approx 3 \times \kappa \times N \times D. \tag{3}$$

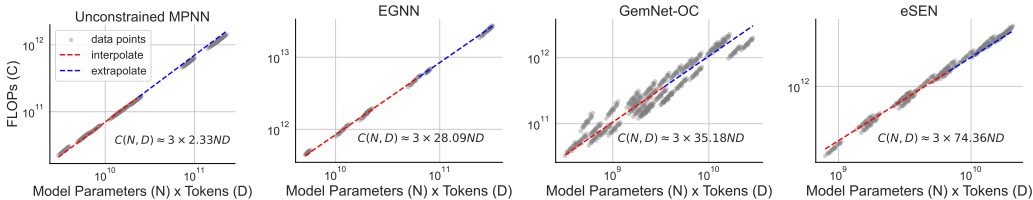

Figure 2: Estimation of $\kappa$ for architectures used in our study.

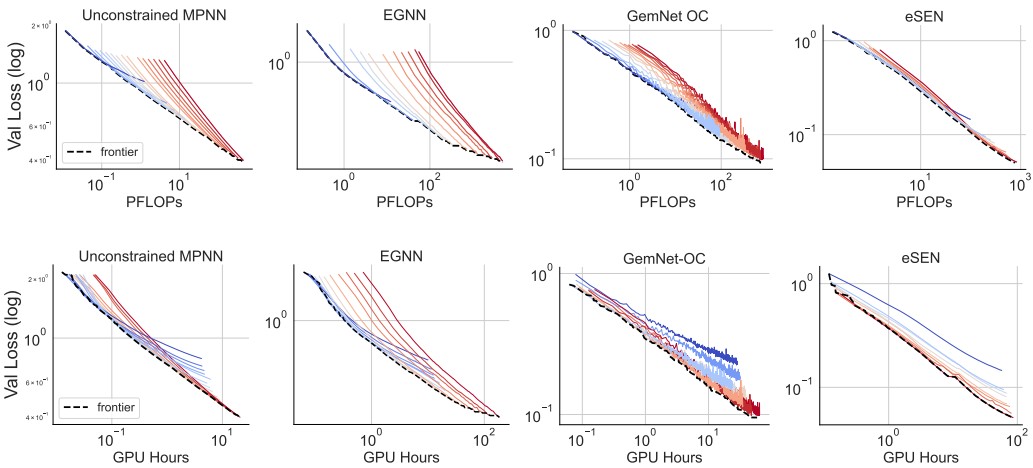

Figure 3: Pareto frontiers of training compute in log–log spaces. **Top**: Efficient loss-FLOPs frontier. **Bottom**: Efficient loss-train-time frontier. Across architectures, the log–log frontiers are approximately linear. Line color encodes model size (small, large).

Here, $\kappa$ is an architecture-dependent constant representing the number of FLOPs required for a single forward pass over one input token. During training, each input incurs both a forward and a backward pass, with the latter approximately doubling the FLOP cost. Consequently, the total training cost per token is $3\kappa$. For architectures dominated by dense linear layers, $\kappa \approx 2$, yielding the widely used expression $C \approx 6ND$ for transformer-based language models.

**Estimating $\kappa$.** We empirically estimate $\kappa$ by varying $N$ and $D$, recording FLOPs for a pass over $D$, and fitting $C$ vs. $ND$. Figure 2 shows a clear linear trend with distinct $\kappa$ per architecture: unconstrained MPNNs with mostly linear layers give $\kappa \approx 2$, whereas equivariant architectures incur larger $\kappa$.

**Compute-Optimal Frontier.** For compute scaling, we follow approach 1 in Hoffmann et al. (2022). For each compute budget, we select the minimum validation loss achieved across runs, yielding the loss–compute Pareto frontiers. As shown in Figure 3 [4], loss-compute frontiers across architectures follows in linear relationships in log-log space. We then fit these frontier points to the power law:

$$L(C) = L_\infty + F_c\, C^{-\gamma_c}, \quad L(H) = L_\infty + F_h H^{-\gamma_h}, \quad (4)$$

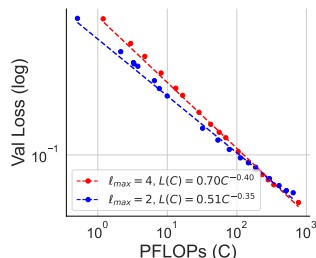

Figure 4: Using higher orders of feature tensors in eSEN leads to better scaling exponents w.r.t compute.

where $L_\infty$ is the irreducible loss for the given architecture and dataset, and $F_C, \gamma_c, F_h$, and $\gamma_h$ are fit parameters. Unlike language modeling with cross-entropy, force-field tasks do not admit a clear theoretical baseline for $L_\infty$ (Brehmer et al., 2025; Wood et al., 2025); therefore, we set $L_\infty \approx 0$ unless noted otherwise. This choice of $L_\infty$ gives exponents that are consistent with the alternative

---

[4]GemNet-OC exhibits noisier learning curves because it relies on empirically estimated per-layer scaling factors—approximated from a few random batches-rather than explicit normalization (e.g., LayerNorm) to control activation variance.

derivation in section 4.3. Figure 1 summarizes our main results, which indicate different exponents for architectures with increasing levels of symmetry expressivity. We also find the argument holds within the same architecture; particularly, fig. 4 shows an improvement in compute-scaling exponents as the max order $\ell_{\max}$ increases from 2 to 4 in eSEN. Finally, it is worth noting that we do not study the effects of denoising pretraining, as done for Orb, a non-equivariant model, by Neumann et al. (2024). Consequently, our compute-scaling results are not directly comparable to this line of work, since their compute budget is allocated differently between pretraining and downstream fine-tuning.

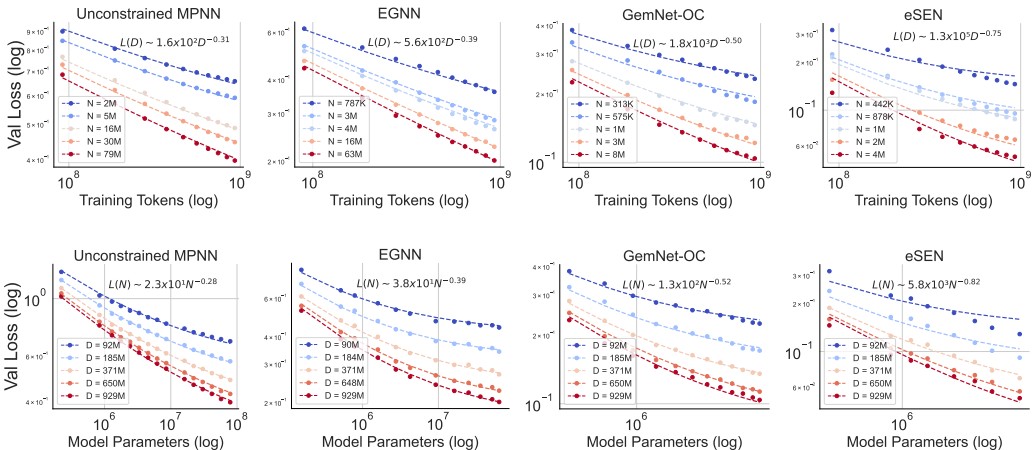

Figure 5: **Top:** Scaling number of training tokens. **Bottom:** Scaling number of parameters

## 4.2 Scaling Parameter and Dataset

**Sum-Power-Law.** To analyze scaling in model size $N$ and dataset size $D$, we follow approach 3 of (Hoffmann et al., 2022) and fit the triplets $(N, D, L)$ to the separable power-law model as:

$$L(N, D) = L_\infty + A \times N^{-\alpha} + B \times D^{-\beta}. \tag{5}$$

$L(N, D)$ is the validation loss represented as a function of $(N, D)$. $L_\infty$ $A, B, \alpha, \beta$ are parameters that we fit. Notably, we found $L_\infty$ to be $\approx 0$ in all architecture families. For each $N$, we measured validation loss at training set fractions $r \in \{0.1, 0.2, \ldots, 1.0\}$, that is $D_r = r \cdot D_{\max}$ [5].

**Scaling Analysis.** Figure 5 presents our fit for four architectures under study. The top row shows the power-law fit in validation loss when the number of training tokens, $D$, is the limiting factor. Power-law exponents $\beta$, are .31, .39, .50 and .75 from left to right. The bottom row shows this relationship when the number of model parameters, $N$, is the bottleneck. Here, the exponent $\alpha$ from left to right is .28, .39, .52, and .82. These results highlight three phenomena:

- **Data Efficiency**: In data-limited scenarios, equivariant models demonstrate superior scaling behaviours compared to unconstrained models, demonstrated by their larger scaling exponents $\beta$. Moreover, equivariance of higher orders translates to larger exponents.

- **Expressivity**: When bottlenecked by model size, equivariant models exhibit higher scaling exponents with respect to $N$. This occurs because explicit symmetry constraints enable greater expressivity with fewer parameters. Furthermore, the scaling exponent gap between high-order architectures (i.e., eSEN, GemNet-OC) and lower-order ones (i.e., EGNN) is considerable. While higher order representations are known to result in better expressivity (Joshi et al., 2023), the fact that the benefit of such representations grows with scale is a novel finding.

- $\alpha \approx \beta$: The exponents remain close across architectures. We discuss this finding in section 4.3.

In brief, we observe larger data-scaling exponents $\beta$ for equivariant networks, consistent with prior reports (Batzner et al., 2022; Brehmer et al., 2025; Wood et al., 2025). Meanwhile, our parameter-scaling exponents $\alpha$ are larger for equivariant networks, and this differs from (Brehmer et al., 2025),

---

[5]Because the GemNet-OC loss curve is high-variance, we smooth it using an exponential moving average with a smoothing factor of 0.9.

which report larger $\alpha$ for unconstrained models; note that the tasks are not directly comparable. Together, the increases in both $\beta$ and $\alpha$ for equivariant models change the slope of the compute-optimal frontier under $C \propto ND$, which is one of our main findings.

### 4.3 COMPUTE-OPTIMAL ALLOCATION

We have presented two scaling laws so far: (1) a power law with respect to the *compute-optimal frontier* in section 4.1, and (2) a *sum-power-law* with respect to parameter count and the number of training tokens in section 4.2. In this section, we discuss the connection between them. Given a fixed compute budget $C$ (FLOPs), we seek the optimal allocation between model size $N$ and training tokens $D$. We pose this as a constrained optimization problem that combines eq. (5) with eq. (3); in particular, we have:

$$N^*(C), D^*(C) = \arg\min L(N, D), \quad 3\kappa ND = C. \tag{6}$$

Recall that $L(N, D) = L_\infty + AN^{-\alpha} + BD^{-\beta}$, with $L_\infty \approx 0$. Let $N^*(C)$ and $D^*(C)$ denote, respectively, the compute-optimal model size and data size for a fixed compute budget $C$. Solving eq. (6) yields $N^*(C) = G\xi^{-a}C^a$, $D^*(C) = G^{-1}\xi^{-b}C^b$, where $\xi = 3\kappa$, $G = \left(\frac{\alpha A}{\beta B}\right)^{\frac{1}{\alpha+\beta}}$, $a = \frac{\beta}{\alpha+\beta}$, $b = \frac{\alpha}{\alpha+\beta}$ (Hoffmann et al., 2022; Brehmer et al., 2025). Furthermore, plugging back the results to $L(N, D)$, we get back the loss-compute frontier power law similar to eq. (4):

$$L(C) = L(N^*(C), D^*(C)) = F_c C^{-\gamma_c}, \tag{7}$$

where $F_c = AG^{-\alpha}\xi^{\gamma} + BG^{\beta}\xi^{\gamma}$, and $\gamma_c = \frac{\alpha\beta}{\alpha+\beta}$. Table 1 presents the values of $F_c$ and $\gamma_c$ obtained from two methods. The results show a good agreement between them, indicating the consistency of our power laws. We further visualize the compute-optimal allocation between model size $N$ and data size $D$ in section 4.3. Across architectures, we find $a \approx b \approx 0.5$, indicating that parameters and tokens should be scaled in roughly equal proportions, consistent with the Chinchilla allocation for transformer language modelling (Hoffmann et al., 2022).

Table 1: Compute-optimal scaling law parameters with $95\%$ confidence intervals; see appendix E. Compute is scaled to PFLOPs.

| Architecture | Param | Fit Method | |
| --- | --- | --- | --- |
| | | Compute-Optimal Frontier eq. (4) | Sum-Power-Law eq. (7) |
| MPNN | $F_c$ | 0.928 [0.925–0.930] | 0.934 [0.863–0.952] |
| | $\gamma_c$ | 0.142 [0.141–0.143] | 0.146 [0.142–0.159] |
| MC-EGNN | $F_c$ | 0.775 [0.761–0.792] | 0.811 [0.784–0.832] |
| | $\gamma_c$ | 0.173 [0.169–0.178] | 0.195 [0.188–0.204] |
| GemNet-OC | $F_c$ | 0.488 [0.485–0.491] | 0.479 [0.424–0.542] |
| | $\gamma_c$ | 0.255 [0.252–0.257] | 0.256 [0.232–0.282] |
| eSEN | $F_c$ | 0.703 [0.696–0.712] | 0.669 [0.575–0.729] |
| | $\gamma_c$ | 0.403 [0.401–0.406] | 0.392 [0.342–0.451] |

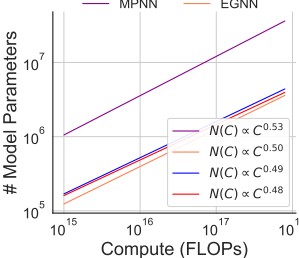
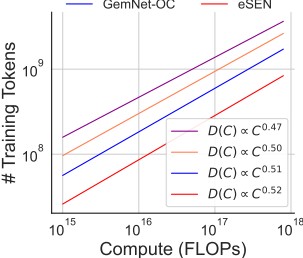

Figure 6: Optimal model size (left) and training size (right) at a fixed level of compute (log-log)

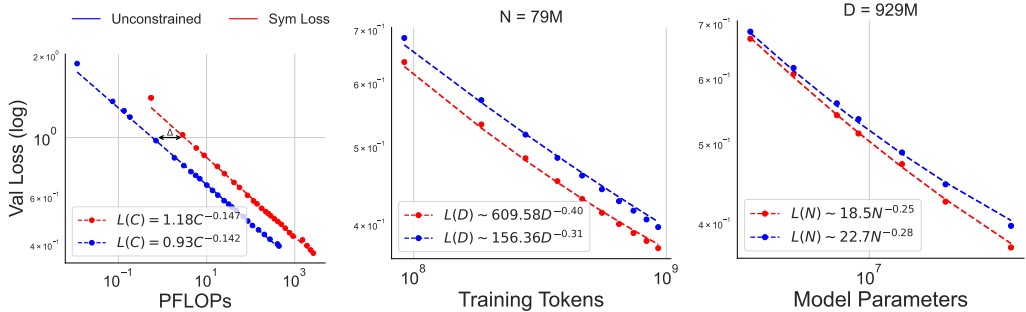

Figure 7: Comparison of scaling exponents for unconstrained MPNNs with/without symmetry regularization. **Left**: Rightward shift of the log–log loss–compute frontier. **Middle**: Symmetry loss increases the data-scaling exponent ($\beta$), indicating improved data efficiency. **Right**: The parameter-scaling exponent ($\alpha$) decreases, suggesting that the regularization benefits larger models more. Validation loss excludes the regularization term (task loss only).

## 4.4 EFFECT OF SYMMETRY LOSS IN SCALING LAWS

In this experiment, we train an unconstrained model augmented with a symmetry-loss term. The loss is $\mathcal{L} = \mathcal{L}_{\text{obj}} + \lambda \mathcal{L}_{\text{sym}}$, where $\mathcal{L}_{\text{obj}}$ is the task loss in eq. (1) and $\mathcal{L}_{\text{sym}}$ is the symmetry loss in eq. (2) wherein we set $M = 5$. We use unit coefficients for both terms (i.e., $\lambda = 1$), as smaller weights ($\lambda \ll 1$) on $\mathcal{L}_{\text{sym}}$ are reported to have negligible effect (Elhag et al., 2025). For validation, we track $\mathcal{L}_{\text{obj}}$, ensuring direct comparison with models trained without the symmetry penalty. We fit the learning-curve trajectories to the functional forms in eq. (4) and eq. (5). Figure 7 shows the resulting fits. Compared with models trained without $\mathcal{L}_{\text{sym}}$, we observe:

- **Opposite Changes in Slopes of $D$ and $N$:** Under the scaling form $L(N, D) = L_\infty + AN^{-\alpha} + BD^{-\beta}$, when $N$ is sufficiently large ($N \to \infty$), adding a symmetry-constraint loss slightly increases the data exponent $\beta$, indicating improved sample-efficiency: the model leverages the regularizer to infer approximate symmetries from data. Conversely, in the infinite-data regime ($D \to \infty$), a smaller model-size exponent $\alpha$ implies that increasing the parameter count $N$ more effectively reduces loss.

- **Unchanged Compute-Optimal Slope:** We hypothesize that because the $N$- and $D$-slopes change in opposite directions, the induced exponent $\gamma$ with respect to compute $C \propto ND$ is preserved. Furthermore, the sampling-based regularizer in eq. (2) functions as data augmentation: in addition to each original sample, we also evaluate $M$ group-transformed inputs and predict the correspondingly transformed targets. As a result, the training FLOPs scale as $C_{\text{sym}} = (M + 1) C_{\text{unconstr}}$, shifting the compute-optimal frontier to the right by $\Delta \approx \gamma \log(M + 1) \approx 0.14 \log_{10}(6)$ in log–log coordinates as shown in fig. 7. Our fits indicate that approximate symmetry enforced via sampling-based augmentation may be unnecessary for compute-optimal scaling, as the relevant scaling exponents remain unchanged.

## 4.5 EFFECT OF MULTI-EPOCH TRAINING IN SCALING LAWS

In the previous sections, we present clear scaling-law trends across architectures in the one-pass training regime, keeping our empirical setup aligned with insights from rigorous theoretical works that focus on this setting (Paquette et al., 2024; Bordelon et al., 2024). Unfortunately, scaling laws for scenarios in which data is repeated for multiple epochs remain under-explored, both in practice and in theory. To our knowledge, Muennighoff et al. (2023) is among the few works investigating this area, showing that under fixed compute, training models for a small number of epochs on repeated data has negligible effects on the loss, behaving almost as if the models are trained on fresh data. We examine this hypothesis in our study by simulating a scenario where data is extremely limited by sampling only $1\%$ of the full dataset, to train the models. The data is repeated over 100 epochs in each training run, which naturally enables the use of data augmentation to improve the data efficiency of unconstrained models. Also, validation losses are recorded after every 1000 gradient steps. Figure 8 shows the resulting fits in this regime. We observe that for unconstrained models trained without augmentation, the loss-compute frontier follows power-law scaling in early epochs, similar

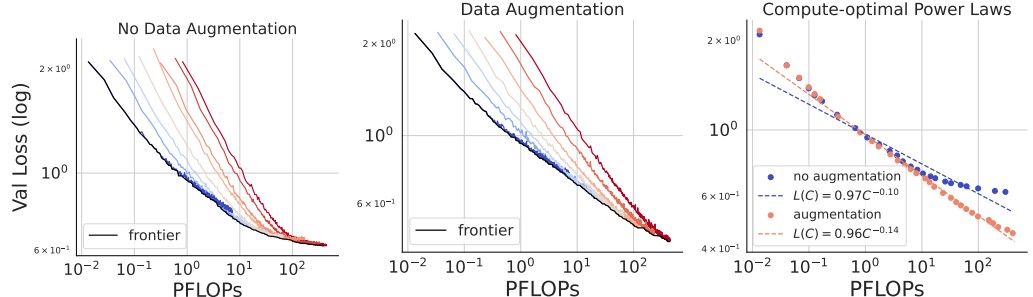

Figure 8: Loss-compute Pareto frontiers for unconstrained models trained on $1\%$ of the data for 100 epochs, shown without data augmentation (**Left**) and with data augmentation (**Middle**). **Right**: The linear trend in log-log space is broken at late training when data augmentation is omitted, and is recovered when data augmentation is utilized.

to findings of Muennighoff et al. (2023). However, this power law breaks down when the number of passes exceeds a certain threshold, as heavy data repetition induces overfitting. In contrast, data augmentation substantially stabilizes the learning curves and ultimately recovers the same power law ($\gamma_c \approx 0.14, F_c \approx 0.96$) with respect to the compute-optimal frontier as in the one-epoch training regime, indicating that the effect of data augmentation under low-data regime is the same as adding fresh data in one-pass training over larger datasets. Furthermore, we also examine the power laws of eSEN under this regime, and observe that its compute-optimal power-law holds as the same in one-pass training. Importantly, fig. 9 reveals that the gap between data augmentation and equivariant networks continues to grow as compute increases through multi-epoch training.

## 5 CONCLUSION, FUTURE WORKS AND LIMITATIONS

Our empirical study of scaling laws in the geometric task of interatomic potentials shows that the degree to which an architecture encodes domain symmetries is correlated with the exponent in its power-law scaling behaviour. The empirical change in exponent is dramatic, suggesting that the role of symmetry potentially extends beyond simply reducing data dimensionality (Sharma & Kaplan, 2022). This is because the degrees of freedom in the input and output are $\approx 3n$ for $n$ atoms, while the rotation group is only three-dimensional. Our findings, therefore, suggest an important future research direction in developing a theory that explains this scaling behaviour. On the practical side, our work provides a recipe for scaling the model and data size in geometric tasks, such as force fields, and it motivates the development of more scalable models that utilize higher-order representations.

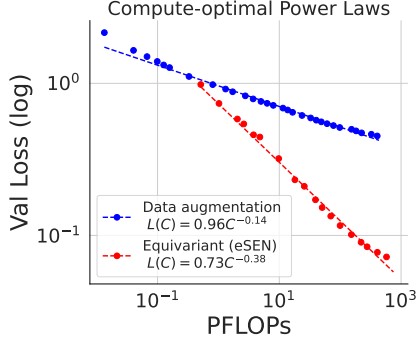

Figure 9: Scaling with training compute for unconstrained models trained with data augmentation and eSEN. The models are trained on $1\%$ of training dataset for 100 epochs.

Other directions for future work are, in part, motivated by the limitations of this work: (1) Our scaling-law analysis focuses on single-epoch, academic-scale settings for NNIPs. Extending it to multi-epoch training and larger models, as well as more diverse models and datasets, is a natural next step. (2) Our study of symmetry losses was confined to one simple choice; it is possible that training with alternative definitions, if scalable, could provide a different scaling behaviour. (3) Our work completely ignores the family of architecture-agnostic equivariant models, such as frame averaging and canonicalization (e.g., Puny et al., 2022; Kaba et al., 2023; Kim et al., 2023b). We plan to study their scaling laws in the future. (4) Finally, we leave to future work a systematic, large-scale evaluation of denoising pretraining for both unconstrained models (Neumann et al., 2024; Rhodes et al., 2025) and equivariant networks (Liao et al., 2024a).

## ACKNOWLEDGMENTS

This research is supported by the Canada CIFAR AI Chairs program, Samsung AI Labs, IVADO, and the NSERC Discovery Grant. Computational resources for experiments are provided by the Digital Research Alliance of Canada (Compute Canada) and Mila.

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

## A  RELATED WORKS

**Neural Scaling Laws.**  Numerous studies investigating the scaling behavior of neural networks (Ahmad & Tesauro, 1988; Henighan et al., 2020; Hoffmann et al., 2022; Kaplan et al., 2020; Sharma & Kaplan, 2022) demonstrate a predictable relationship: performance improves as model size $N$, dataset size $D$, and computational budget $C$ increase. Various functional forms have been proposed to model these scaling laws. Using test error $\epsilon$ as the evaluation metric, Cortes et al. (1993) and Hestness et al. (2017) proposed the functional form $L = ax^{-b} + L_{\infty}$, where $b > 0$, $L_{\infty} \geq 0$ denotes the irreducible error, and $x$ can be $N$, $D$, or $C$. While useful, this form may result in infinite error as the scaling variable $x$ approaches zero (e.g., for models performing random guessing in classification tasks). To address this limitation, Zhai et al. (2022) introduced a more general form, $L = a(x + c)^{-b} + L_{\infty}$, where the parameter $c$ represents an effective offset, indicating the scale at which performance significantly surpasses random guessing. The coefficients of these power laws have been empirically explored across various research domains and tasks, including autoregressive generative modeling (Henighan et al., 2020; Kaplan et al., 2020; Hoffmann et al., 2022) and computer vision (Zhai et al., 2022; Henighan et al., 2020; Alabdulmohsin et al., 2022). Further research by Hoffmann et al. (2022) and Snell et al. (2025) has improved methodologies for studying scaling laws, allowing for the determination of optimal scaling strategies to achieve the best performance on specific tasks under given constraints. Additionally, Caballero et al. (2023) introduced "broken" scaling laws to better model and extrapolate neural network scaling behaviours, particularly when the scaling functions exhibit non-monotonic transitions.

**Scaling laws for MPNNs on Molecular Graphs.**  Learning accurate molecular representations is a fundamental challenge in drug discovery and computational chemistry. Numerous studies have investigated the scalability of graph neural networks (GNNs), particularly message-passing neural networks (MPNNs), for predicting molecular properties. For 2D molecular graphs, prior research by Liu et al. (2024); Sypetkowski et al. (2024); Li et al. (2025) has demonstrated the promising scalability of MPNNs, showing that network performance follows a power-law scaling behaviour with increases in both dataset and model sizes. Similar scaling trends have also been observed by Frey et al. (2023); Li et al. (2025); Wood et al. (2025) for $E(3)/SE(3)$ equivariant MPNNs trained on 3D atomistic systems. In contrast to these observations, Pengmei et al. (2025) demonstrate that scaling behaviours of geometric GNNs deviate from conventional power laws across different settings, including self-supervised, supervised, and unsupervised learning.

Our work extends scaling-law analysis across NNIP architectures. Unlike Wood et al. (2025), which derives compute-optimal scaling for equivariant models on mixed materials–molecule datasets with periodic coordinates and other auxiliaries, we focus on molecules using only atomic 3D coordinates and atomic numbers. Despite using the eSEN same backbone, Wood et al. (2025) report that, for dense models [6], the compute-optimal strategy scales model size $N$ faster than data size $D$, whereas in our setting we observe nearly equal scaling between $N$ and $D$; though the tasks are different. Relative to Frey et al. (2023), our study uses a substantially larger force-field dataset, enabling more robust scaling estimates that consider different architectures.

## B  SUPPORTING FIGURES

## C  DESIGN PRINCIPLES FOR GEOMETRIC MESSAGE PASSING

Table 2: Architectures and their expressivity.

| Architectures | Characteristic | Tensor Order ($\ell$) | Body Order ($\nu$) |
|---|---|---|---|
| Unconstrained MPNN | unconstrained | - | 2 |
| GemNet-OC | invariant | 0 | 4 |
| EGNN | equivariant | 1 | 2 |
| eSEN | equivariant | $\geq 2$ | 2 |

---

[6]The authors also study the effect of linear mixture-of-experts, while we don't consider this in our work.

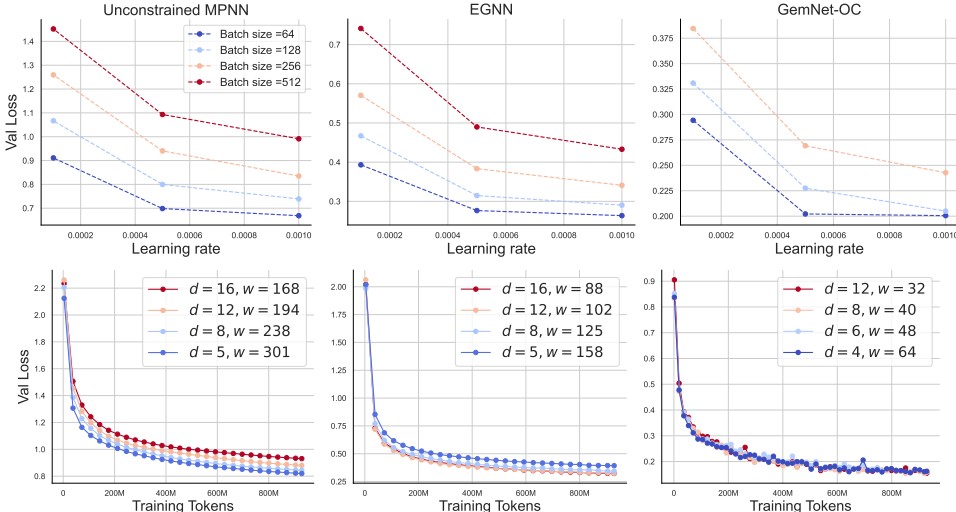

Figure 10: A sweep over batch size-learning rate (top row), and depth-width (bottom row) for three architectures with numbers of parameters are approximately equal to $1M$.

Following Joshi et al. (2023), we categorize the models and their expressivity in Table 2. The tensor order $\ell$ denotes the order of the geometric tensor embeddings processed by each model. The body order refers to the number of nodes participating in a message function. Because GemNet-OC incorporates dihedral-angle information, its two-hop messages depend simultaneously on the states of four nodes, and thus it is classified as four-body. By contrast, the remaining architectures use one-hop message passing that depends only on the source and target nodes, i.e., two-body.

Let $\{h_1^{(t)}, \ldots, h_n^{(t)}\}$ denote the node embeddings at layer $t$, with $h_i^{(0)}$ initialized from input features $z_i$. For a distance cutoff $c > 0$, define the neighborhood

$$\mathcal{N}(v) \;=\; \{\, u \neq v \mid \|x_u - x_v\| \leq c \,\}.$$

We update node $v$ by aggregating messages from its neighbors using a permutation-invariant (mean) aggregator, making the overall layer permutation-equivariant:

$$h_v^{(t+1)} \;=\; \frac{1}{|\mathcal{N}(v)|} \sum_{u \in \mathcal{N}(v)} m_{u \to v}^{(t)}, \qquad m_{u \to v}^{(t)} \;=\; \phi_m^{(t)}(\cdot), \tag{8}$$

Let $r_{uv} := x_u - x_v$ be the relative position vector and $w := \dim h_v^{(t)}$ be the embedding width. The symmetry properties (e.g., $E(3)/SE(3)$ invariance or equivariance) are determined by architectural choices in $\phi_m^{(t)}$, in particular, how it uses $r_{uv}$ (e.g., through rotational invariants such as $\|r_{uv}\|$ or via equivariant tensor constructions). We discuss specific message constructions in the next section.

## C.1 UNCONSTRAINED MESSAGE PASSING

Following Duval et al. (2023), geometry is injected directly:

$$m_{u \to v}^{(t)} = \phi_m\big(h_u^{(t)}, h_v^{(t)}, r_{uv}, \|r_{uv}\|_2\big), \tag{9}$$

with $h_u^{(t)}, h_v^{(t)} \in \mathbb{R}^w$ and $\phi_m : \mathbb{R}^{2w+4} \to \mathbb{R}^w$ is an MLP. Because raw vectors $r_{uv}$ are processed without symmetry constraints, rotational equivariance is not enforced, and thus the node embeddings $h_v^{(t)}$ are not rotationally invariant.

## C.2 DIRECTIONAL MESSAGE PASSING

GemNet-T (Klicpera et al., 2021) and GemNet-OC (Gasteiger et al., 2022) construct messages from multi–body $E(3)$-invariant geometric features, including pairwise distances, bond (three-body) angles, and dihedral (four-body) angles. Define the bond angle $\varphi_{uvk} := \angle\big(r_{vu}, r_{vk}\big)$ and the dihedral

angle $\omega_{uvkj}$ as the angle between the planes $(u, v, k)$ and $(v, k, j)$ (e.g., via normals $n_1 \propto r_{vu} \times r_{vk}$ and $n_2 \propto r_{kj} \times r_{kv}$). Then the message from $u$ to $v$ is

$$m_{u \to v}^{(t)} = \sum_{\substack{k \in \mathcal{N}(v) \setminus \{u\} \\ j \in \mathcal{N}(k) \setminus \{u,v\}}} \phi_m\Big(h_u^{(t)}, h_v^{(t)}, \|r_{uv}\|_2, \varphi_{uvk}, \omega_{uvkj}\Big), \qquad (10)$$

where $h_u^{(t)}, h_v^{(t)} \in \mathbb{R}^w$ are scalar node embeddings and $\phi_m$ is a learnable function operating on $E(3)$-invariant inputs (distance, angles) together with scalar features. Because the geometric inputs are $E(3)$-invariant (rotation/translation invariant) and $h$'s are scalar channels, the resulting message is $E(3)$-invariant as well. Linear and bilinear interactions inside $\phi_m$ do not affect this invariance so long as they act on invariant/scalar quantities.

## C.3 CARTESIAN-VECTOR MESSAGE PASSING

The original EGNN uses a single vector channel (node coordinates) (Satorras et al., 2021), which limits expressive power (Joshi et al., 2023). To address this, we use a multi-channel extension, MC-EGNN (Levy et al., 2023). Each atom $v$ carries both scalar features $h_v^{(t)} \in \mathbb{R}^w$ and $E$ vector channels $X_v^{(t)} \in \mathbb{R}^{3 \times E}$, with the relative vector $X_{uv}^{(t)} := X_u^{(t)} - X_v^{(t)} \in \mathbb{R}^{3 \times E}$.

MC-EGNN maintains invariance/equivariance by updating the invariant node embeddings $h_v$ and the multi-channel equivariant vectors $X_v$ using invariant messages. In particular, messages depend only on rotation–translation invariants:

$$m_{u \to v}^{(t)} = \phi_m\big(h_u^{(t)}, h_v^{(t)}, \|X_{uv}^{(t)}\|_e^2\big), \qquad \phi_m : \mathbb{R}^{2w+E} \to \mathbb{R}^w, \qquad (11)$$

where $\|X_{uv}\|_e \in \mathbb{R}^E$ denotes the channel-wise Euclidean norm (applied over the 3 spatial components). The invariant message $m_{u \to v}^{(t)}$ from Equation (11) is then used in Equation (8) to update $h_v^{(t+1)}$. In addition, MC-EGNN updates the vectors via a channel mixer:

$$X_v^{(t+1)} = X_v^{(t)} + \frac{1}{|\mathcal{N}(v)|} \sum_{u \in \mathcal{N}(v)} \frac{1}{E} X_{uv}^{(t)} \Phi_x\big(m_{u \to v}^{(t)}\big), \qquad (12)$$

where $\Phi_x$ is a linear map with weights $W_x : \mathbb{R}^w \to \mathbb{R}^{E \times E'}$ followed by a reshape. This channel mixing preserves equivariance because $E(3)$ actions (and permutations) act on the 3D indices but *not* on the channel index.

$\Theta(1)$**-Variance Scaling.** Our empirical analyses suggest that increasing $E$ significantly improves performance; see appendix F. This, in turn, suggests that studying scaling laws requires scaling both the invariant dimension $w$ and the equivariant channels $E$. However, very large $E$ can be computationally expensive, and scaling arbitrarily may cause exploding gradients due to the matrix-valued function $\Phi_x$ and the matrix product $X_{uv}\Phi_x(m_{u \to v})$. To scale properly, one should ensure that each layer's output scales as $\Theta(1)$ and gradient update scales as $\Theta(1)$ (Yang et al., 2021), i.e., are invariant across widths. Let $W_x \sim \mathcal{N}(0, \sigma^2)$ with $\sigma = \Theta(\sqrt{\min(w, EE')/w^2})$. Setting $E = E' \approx \sqrt{w}$ yields $\sigma \approx \Theta(1/\sqrt{w})$. Under $\mu$P, we assume the entries of $m_{u \to v}$ and $X_{uv}$ have variance $\Theta(1)$; then $W_x m_{u \to v}$ also has $\Theta(1)$-entries, and since *reshape* has no parameters, the entries of $\Phi_x(m_{u \to v})$ remain $\Theta(1)$. For the matrix product, because both $X_{uv}$ and $\Phi_x(m_{u \to v})$ have $\Theta(1)$-variance entries, $(X_{uv}\Phi_x(m_{u \to v}))_{3 \times E}$, which sums over $E$, scales as $\Theta(E)$. To keep gradient updates stable across widths, we scale $X_{uv}\Phi_x(m_{u \to v})$ by $1/E \approx 1/\sqrt{w}$, instead of $1/\sqrt{E}$, which has the same effect as scaling the logits of dot-product attention by $1/\text{emb\_dim}$ rather than $1/\sqrt{\text{embed\_dim}}$ as discussed by Yang et al. (2021).

## C.4 HIGH-ORDER TENSOR MESSAGE PASSING

**Irreducible Representations.** High-order equivariant models (e.g., Thomas et al. (2018); Anderson et al. (2019); Batzner et al. (2022); Batatia et al. (2022); Liao et al. (2024b); Wood et al. (2025)) use $SO(3)$ irreducible representations (irreps) as node embeddings:

$$h_u^{(t)} = \bigoplus_{\ell=0}^{\ell_{\max}} h_{u,\ell}^{(t)}, \qquad h_{u,\ell}^{(t)} \in \mathbb{R}^{C_\ell} \otimes \mathbb{V}^{(\ell)}, \quad \dim \mathbb{V}^{(\ell)} = 2\ell + 1, \qquad (13)$$

with $\bigoplus$ denotes concatenation of multiple order-$\ell$ tensors that are expanded by $C_\ell$ channels, and thus total dimension is $d := \sum_{\ell=0}^{\ell_{\max}} C_\ell(2\ell+1)$. Assume all orders have the same number of channels $C$, the atom embedding has a size of $w = C(\ell_{\max}+1)^2$.

**SO(3) Convolution.** Let $\hat{r}_{uv} = r_{uv}/\|r_{uv}\|$. Messages couple source irreps with spherical harmonics:

$$m_{u \to v, \ell_3}^{(t)} = \sum_{\ell_1, \ell_2} w_{\ell_1 \ell_2 \ell_3} \bigoplus_{m_3} \sum_{m_1, m_2} h_{u,(\ell_1,m_1)}^{(t)} C_{(\ell_1,m_1),(\ell_2 m_2)}^{(\ell_3 m_3)} Y_{\ell_2}^{m_2}(\hat{r}_{uv}) \tag{14}$$

where $C_{\ell_1 m_1, \ell_2 m_2}^{\ell_3 m_3}$ are Clebsch–Gordan coefficients, $w_{\ell_1, \ell_2, \ell_3}$ are learnable weights, and $Y_\ell$ is the order-$\ell$ spherical harmonics of the unit displacement vector $\hat{r}_{uv}$, $|\ell_1 - \ell_3| \le \ell_2 \le |\ell_1 + \ell_3|$ and $m_i \in \{-\ell_i, \ldots, \ell_i\}$.

**Efficient Convolution.** eSCN/eSEN (Passaro & Zitnick, 2023; Fu et al., 2025) sparsify eq. (14) by rotating vector $r_{uv}$ into an edge-aligned frame. Let $R_{uv} \in \mathbb{R}^{3 \times 3}$ such that $R_{uv}\hat{r}_{uv} = (0, 1, 0)$. Then $Y_{\ell_2}^{m_2}(R_{uv}\hat{r}_{uv}) = 0$ unless $m_2 = 0$. Therefore, we can simplify eq. (14) as:

$$m_{u \to v, \ell_3}^{(t)} = D_{\ell_3}^{-1} \sum_{\ell_1, \ell_2} w_{\ell_1, \ell_2, \ell_3} \bigoplus_{m_3} \sum_{m_1} \tilde{h}_{u,(\ell_1,m_1)}^{(t)} C_{(\ell_1,m_1),(\ell_2,0)}^{(\ell_3,m_3)}, \tag{15}$$

here $\tilde{h}_{(\ell_1,m_1)}^{(t)} = D_{\ell_1} h_{(\ell_1,m_1)}^{(t)}$ where we denote $D_{\ell_1} := D_{\ell_1}(R_{uv})$ and $D_{\ell_3} := D_{\ell_3}(R_{uv})$ denote Wigner-D matrices of order $\ell_1$ and $\ell_3$, respectively. The output is rotated back by $D_{\ell_3}$ to ensure equivariance, and without loss of generality, we re-scale $Y_{\ell_2}^{m_2}(R_{uv}\hat{r}_{uv})$ to 1. Given that $m_2 = 0$, $C_{(\ell_1,m_1),(\ell_2,0)}^{(\ell_3,m_3)}$ are non-zero only when $m_1 = \pm m_3$. This further simplifies the computation to:

$$m_{u \to v, \ell_3}^{(t)} = D_{\ell_3}^{-1} \sum_{\ell_1, \ell_2} w_{\ell_1, \ell_2, \ell_3} \bigoplus_{m_3} \left( \tilde{h}_{v,(\ell_1,m_3)}^{(t)} C_{(\ell_1,m_3),(\ell_2,0)}^{(\ell_3,m_3)} + \tilde{h}_{v,(\ell_1,-m_3)}^{(t)} C_{(\ell_1,-m_3),(\ell_2,0)}^{(\ell_3,m_3)} \right). \tag{16}$$

Rearranging eq. (16), we obtain:

$$m_{u \to v, \ell_3}^{(t)} = D_{\ell_3}^{-1} \sum_{\ell_1} \bigoplus_{m_3} \left( \tilde{h}_{v,(\ell_1,m_3)}^{(t)} \sum_{\ell_2} w_{\ell_1, \ell_2, \ell_3} C_{(\ell_1,m_3),(\ell_2,0)}^{(\ell_3,m_3)} + \tilde{h}_{v,(\ell_1,-m_3)}^{(t)} \sum_{\ell_2} w_{\ell_1, \ell_2, \ell_3} C_{(\ell_1,-m_3),(\ell_2,0)}^{(\ell_3,m_3)} \right).$$
$$\tag{17}$$

Passaro & Zitnick (2023) propose to replace the Clesbh-Gordon coefficients with parameterized weights as:

$$\tilde{w}_{m_3}^{(\ell_1,\ell_3)} = \sum_{\ell_2} w_{\ell_1, \ell_2, \ell_3} C_{(\ell_1,m_3),(\ell_2,0)}^{(\ell_3,m_3)} = \sum_{\ell_2} w_{\ell_1, \ell_2, \ell_3} C_{(\ell_1,-m_3),(\ell_2,0)}^{(\ell_3,-m_3)}, \quad \text{for } m \ge 0, \tag{18}$$

$$\tilde{w}_{-m_3}^{(\ell_1,\ell_3)} = \sum_{\ell_2} w_{\ell_1, \ell_2, \ell_3} C_{(\ell_1,-m_3),(\ell_2,0)}^{(\ell_3,m_3)} = -\sum_{\ell_2} w_{\ell_1, \ell_2, \ell_3} C_{(\ell_1,m_3),(\ell_2,0)}^{(\ell_3,-m_3)}, \quad \text{for } m < 0. \tag{19}$$

Plugging back this into eq. (16), we obtain:

$$m_{u \to v, \ell_3}^{(t)} = D_{\ell_3}^{-1} \sum_{\ell_1} y_{\ell_3}^{(\ell_1)}, \tag{20}$$

where:

$$y_{\ell_3,0}^{(\ell_1)} = \tilde{w}_0^{(\ell_1,\ell_3)} \tilde{h}_{u,(\ell_1,0)}^{(t)} \tag{21}$$

$$\begin{pmatrix} y_{(\ell_3,m_3)}^{(\ell_1)} \\ y_{(\ell_3,-m_3)}^{(\ell_1)} \end{pmatrix} = \begin{pmatrix} \tilde{w}_{m_3}^{(\ell_1,\ell_3)} & -\tilde{w}_{-m_3}^{(\ell_1,\ell_3)} \\ \tilde{w}_{-m_3}^{(\ell_1,\ell_3)} & \tilde{w}_{m_3}^{(\ell_1,\ell_3)} \end{pmatrix} \cdot \begin{pmatrix} \tilde{h}_{u,(\ell_1,m_3)}^{(t)} \\ \tilde{h}_{u,(\ell_1,-m_3)}^{(t)} \end{pmatrix} \quad \text{for } m_3 > 0. \tag{22}$$

Therefore, the overall computation is reduced to an equivalent $SO(2)$ linear operation with a parameterized kernel as in eq. (22). In eSEN, SO(2) blocks are applied only for values $|m_3| \le m_{max} \le \ell_{max}$; we set $m_{max} = 2$ as similar to the default setting.

Table 3: Scaling ladder for unconstrained MPNN.

| Depth | Width | # Params |
|---|---|---|
| 5 | 128 | 222404 |
| 5 | 256 | 838020 |
| 5 | 320 | 1293284 |
| 5 | 375 | 1763133 |
| 5 | 441 | 2422704 |
| 5 | 517 | 3311714 |
| 5 | 607 | 4543769 |
| 5 | 712 | 6226800 |
| 5 | 835 | 8535023 |
| 5 | 1150 | 16101729 |
| 5 | 1349 | 22109506 |
| 5 | 1583 | 30389712 |
| 5 | 1857 | 41755644 |
| 5 | 2179 | 57415632 |
| 5 | 2557 | 78974296 |

Table 4: Scaling ladder for EGNN. We scale the number of channels for equivariant vectors as $\sqrt{w}$, ensuring stable update and $\Theta(1)$-variance entries when scaling $w$.

| Depth $d$ | Width $w$ | # Params |
|---|---|---|
| 12 | 32 | 156358 |
| 12 | 48 | 306199 |
| 12 | 56 | 403992 |
| 12 | 64 | 516553 |
| 12 | 81 | 786925 |
| 12 | 102 | 1195553 |
| 12 | 120 | 1590131 |
| 12 | 160 | 2745453 |
| 12 | 200 | 4231015 |
| 12 | 300 | 9214218 |
| 12 | 500 | 24959524 |
| 12 | 600 | 35673024 |
| 12 | 721 | 51092972 |
| 12 | 800 | 63035228 |

## D EXPERIMENTAL SETUP

**Graph Construction.** Except for GemNet-OC, geometric graphs are built with a radial cutoff of 6 Å and a maximum of 30 neighbors per atom. For GemNet-OC, using the same cutoff occasionally failed on molecules where its high-order message passing requires a larger candidate neighborhood. Accordingly, we use a 10 Å cutoff and cap the neighborhood at 50 neighbors for this architecture.

**Direct-Force Prediction.** Although the pre-training performance of direct-force models does not always transfer seamlessly to certain downstream physical tasks, these models provide substantial efficiency gains during pre-training. We focus on the scaling behaviour of direct-force models in the pre-training regime, i.e., training across large collections of molecular systems, and therefore employ them consistently throughout our experiments.

**Energy Reference and Normalization.** Following Wood et al. (2025)[7], we apply the same reference scheme to the energies. Then, we normalize those referenced energies as $e' = (e - \mu)/\sigma$, where $\mu$ and $\sigma$ are the sample mean and standard deviation estimated from the training set. Atomic

---

[7]https://github.com/facebookresearch/fairchem/tree/main

forces for atom $i$, $f_i' = -\partial e/\partial x_i$, are invariant to constant energy shifts; accordingly, we rescale them by the same factor: $f_i' = f_i/\sigma$.

Table 5: Scaling ladder for eSEN. Since eSEN use high-order tensors expanded into $C$ channels each, the embedding width $w = (\ell_{\max} + 1)^2 C$.

| Depth | $\ell_{\max}$ | $m_{\max}$ | # Channels $C$ | # Hidden Channels | # Params |
|---|---|---|---|---|---|
| 12 | 4 | 2 | 4 | 32 | 441778 |
| 12 | 4 | 2 | 8 | 32 | 878434 |
| 12 | 4 | 2 | 10 | 32 | 1100746 |
| 12 | 4 | 2 | 12 | 32 | 1325714 |
| 12 | 4 | 2 | 16 | 32 | 1783618 |
| 12 | 4 | 2 | 20 | 32 | 2252146 |
| 12 | 4 | 2 | 32 | 32 | 3721474 |

Table 6: Scaling ladder for GemNet-OC.

| Depth | Width | # Params |
|---|---|---|
| 4 | 32 | 312992 |
| 4 | 48 | 575472 |
| 4 | 64 | 937280 |
| 4 | 80 | 1398416 |
| 4 | 96 | 1958880 |
| 4 | 112 | 2618672 |
| 4 | 128 | 3377792 |
| 4 | 144 | 4236240 |
| 4 | 160 | 5194016 |
| 4 | 176 | 6251120 |
| 4 | 192 | 7407552 |
| 4 | 204 | 8340060 |

**Training Hardware.** We trained all models across the different architectural families under identical hardware conditions on NVIDIA 40GB A100 GPUs of the same compute cluster - each run within a single unit. This setup naturally incorporates overheads such as data loading, CPU bottlenecks, and metrics logging. During training, we continuously monitored validation losses along with the corresponding wall-clock training times. The reported training times thus include forward and backward passes over training samples, as well as forward passes over validation samples during intermediate evaluation checkpoints. We chose batch sizes to strike a balance between performance and computational efficiency, as using extremely small batch sizes is impractical. Based on the empirical results in Figure 10, we set the batch size to 128 for EGNN, and 64 for the remaining architectures [8]. These batch sizes were fixed across model sizes within each model family.

**Scaling Ladder.** Tables 3, 4, 5, 6 detail model sizes of all architectures used in this study.

Table 7: Scaling parameters for *sum-power-law in eq.* (5) with $95\%$ confidence intervals.

| Architecture | $\log_{10}(A)$ | $\log_{10}(B)$ | $\alpha$ | $\beta$ |
|---|---|---|---|---|
| Unconstrained MPNN | 1.356 [1.307-1.371] | 2.194 [2.147-2.608] | 0.276 [0.266- 0.278] | 0.311 [0.301-0.368] |
| EGNN | 1.582 [1.494,1.692] | 2.750 [2.660-2.863] | 0.387 [0.370-0.408] | 0.394 [0.382-0.401] |
| GemNet-OC | 2.109 [1.901-2.422] | 3.261 [2.842-3.607] | 0.524 [0.484-0.584] | 0.499 [0.444-0.544] |
| eSEN | 3.760 [3.119-4.333] | 5.129 [4.348-6.224] | 0.817 [0.706-0.918] | 0.753 [0.662-0.888] |

---

[8]For fair comparison, we double EGNN's reported training time.

## E  UNCERTAINTY IN SCALING LAWS.

Due to the cost of training NNIPs, we perform scaling study within a range of compute and model sizes and do not consider tuning other hyper-parameters, such as weight decay. Thus, we construct 95% confidence intervals on the fit parameters of eq. (4) and eq. (5) from 1000 non-parametric bootstraps. Table 7 demonstrates the values of fit parameters of eq. (5) along with confidence intervals shown in parenthesis.

## F  EFFECT OF SCALING VECTOR CHANNELS

We evaluate the effect of multi-channel vectors in EGNN by plotting validation loss against the number of vector channels $E$ in fig. 11. Loss consistently decreases as $E$ grows.

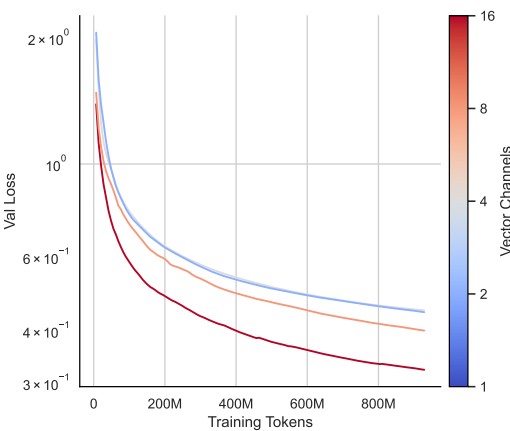

Figure 11: Effect of scaling number of equivariant channels in EGNN

## G  SYMMETRY ADVANTAGES ON DATASETS OF DIVERSE MOLECULAR TYPES

In this experiment, we investigate whether the advantages of symmetry diminish when models are trained on datasets containing more diverse molecular types, e.g., electrolytes, metal complexes, and biomolecules rather than only neutral species. We train an unconstrained MPNN and eSEN on the 4M split of the OpenMol dataset, which is sampled uniformly across the diverse molecular types mentioned above. For validation, we use a held-out subset of 79K samples from the entire 4M split, using the remainder for training. As shown in fig. 12, we observe that the benefits gained by symmetry-aware design still persist even with this highly diverse dataset. Specifically, the difference in scaling exponents between the unconstrained MPNN and eSEN remains significant, a result similar to our findings for the neutral species split. It is important to note that the scaling exponents for each architecture itself may differ from those observed in the neutral split. This is expected because the coefficients of neural scaling laws are also dependent on the training dataset (Maloney et al., 2022; Bahri et al., 2024; Bordelon et al., 2024).

## H  EFFECT OF TEST-TIME AUGMENTATION IN SCALING LAWS

An unconstrained model $\phi_\theta$ can be made equivariant at test time via group averaging (GA) with no additional training cost.

$$f_\theta(x) = \frac{1}{M} \sum_{i=1}^{M} \rho_{\text{out}}(g_i^{-1})\phi_\theta(\rho_{\text{in}}(g_i)x),\tag{23}$$

where $g_i \sim \mu_G$ is a sample from the Haar measure, $\rho_{\text{in}}$ and $\rho_{\text{out}}$ denote linear actions on input and output of $\phi_\theta$, respectively. In this section, we explore the effect of GA on the performance of

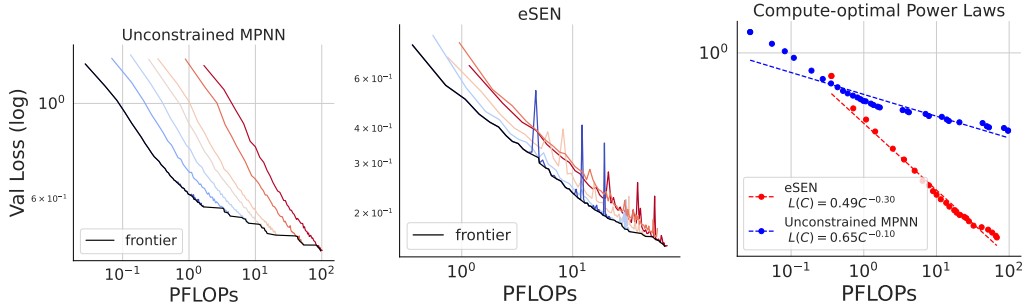

Figure 12: Loss-compute Pareto frontiers for unconstrained models (**Left**) and eSEN (**Middle**) trained on OMol-4M split. **Right**: The compute-optimal power laws reveal that the benefits of symmetry persist at scale, even when models are trained on datasets containing a high diversity of molecular types.

unconstrained models at scale, especially when the model's parameter count and number of group elements increase.

Figure 13 shows that group averaging yields only a minimal improvement in the performance of $\phi_\theta$ across all model sizes. Furthermore, this improvement saturates as the number of rotations, $M$, increases beyond a certain threshold. To better understand this behavior, we fit scaling laws with respect to the number of parameters $N$ for both the baseline and the GA models:

$$L - L_D = AN^{-\alpha}. \tag{24}$$

We use the parameter scaling relationship to analyze the impact of group averaging (GA). $L_D$ represents the lowest achievable loss for $\phi_\theta$ at a fixed data size $D$; see eq. (7). For this analysis, we utilize models trained on the largest dataset, $D_{\max} \approx 9.2 \times 10^8$ (atoms). Based on the results in section 4, the data-limited loss is $L_D \approx 1.6 \times 10^{-2} \times D_{\max} \approx 0.223$. We consider the GA performance at the onset of saturation, specifically at $M = 32$. The fit results in fig. 13 show that group averaging preserves the power-law exponent $\alpha$ that governs the scaling of $\phi_\theta$. As the parameter count increases, the performance shows only a slight downward shift in the log-log scaling relationship, due to a minor change in the multiplying constant $A$.

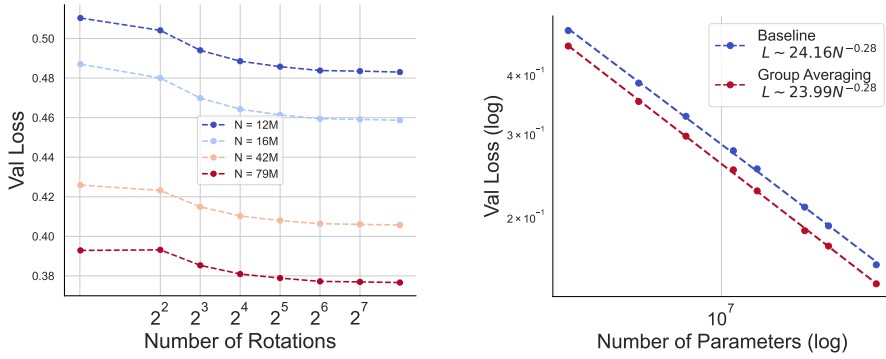

Figure 13: Scaling analysis for group averaging (GA) at test time. **Left**: The benefit derived from increasing the number of rotations saturates beyond a certain threshold. **Right**: Scaling performance with respect to parameter count shows that utilizing group averaging results in a slight downward shift of the linear trend in log-log space, while the critical scaling exponent remains unchanged.

# I    DECOMPOSING ENERGY AND FORCE LOSS

Figure 14 shows the decomposition of the total loss in eq. (1) into its energy and force components. We observe that the energy learning curves are relatively noisy, whereas the force losses remain smooth across model sizes for each architecture.

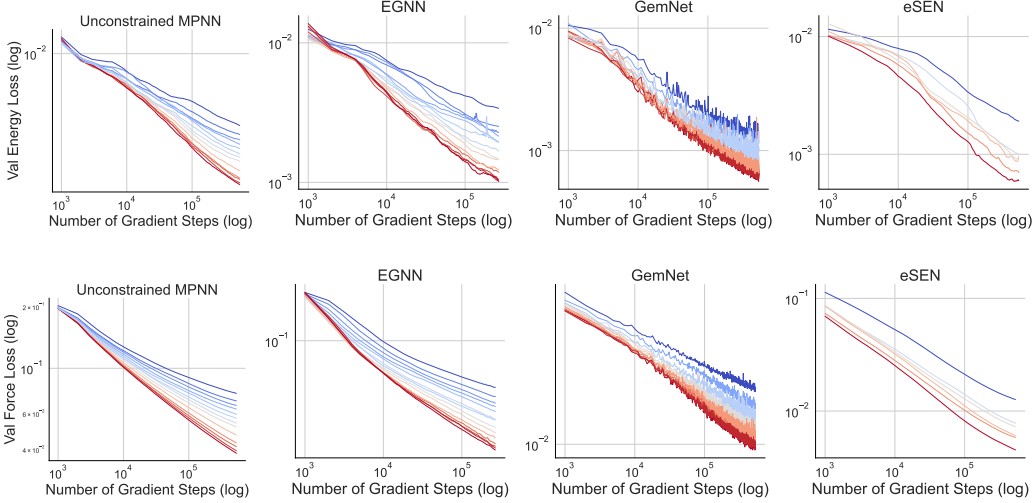

Figure 14: Learning curves of all models, including Energy Loss (**Top**) and Force Loss (**Bottom**). Line color encodes model size (small, large)
.

# J    EFFECT OF TRANSLATION INVARIANCE

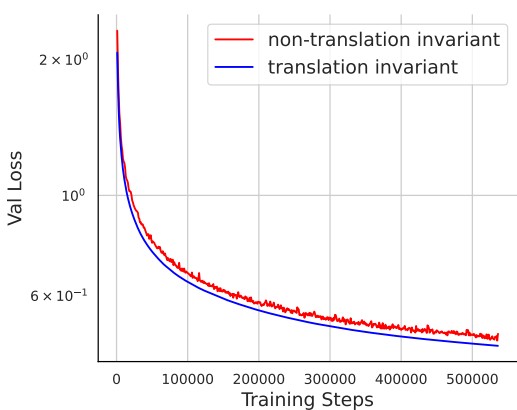

Figure 15: MPNNs exhibit a performance drop when translation invariance is not maintained.

# K    TRAINING INSTABILITIES OF VANILLA TRANSFORMERS

We further push the limit of lacking inductive biases by training a vanilla transformer for force-field tasks. In particular, we train a GPT-style encoder in which atomic coordinates are fed directly into the model by concatenating them with embeddings computed from atomic numbers. As a result, E(3) equivariance is completely ignored in this setup. Moreover, because the transformer learns global attention over all atom nodes, the inductive bias of local neighborhoods, which is critical in NNIPs, is also abandoned. As shown in Figure 16, vanilla transformers fail to exhibit meaningful learning, as the learning curves saturate rapidly and remain unchanged for the remainder of training.

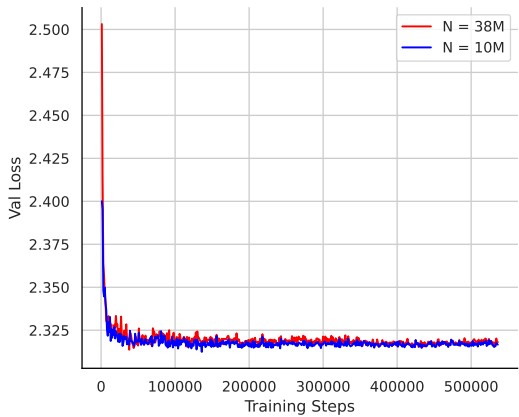

Figure 16: Training instabilities of vanilla transformers on force-field tasks.

## L    USAGE OF LARGE LANGUAGE MODELS

We used LLMs to assist with the writing of the paper (mainly for polishing) and also for coding.

