# OpenReview forum: "Scaling Laws and Symmetry, Evidence from Neural Force Fields"
_ICLR.cc/2026/Conference — ICLR 2026 Poster_

### Official Review · Reviewer_RXkz · 2025-10-30

**Soundness:** 2
**Presentation:** 3
**Contribution:** 2
**Rating:** 4
**Confidence:** 4

**Summary:**

This paper presents an empirical investigation of scaling laws for neural force fields used in learning interatomic potentials. The authors systematically study how equivariant and non-equivariant architectures scale with respect to data size, model parameters, and compute budget. The key finding is that equivariant architectures exhibit superior scaling exponents compared to non-equivariant models, with higher-order equivariant representations showing the best scaling behavior. The study provides power-law fits demonstrating "architecture-dependent exponents" and suggests that compute-optimal training requires scaling data and model size together. The authors argue that fundamental inductive biases like symmetry should be built into architectures rather than left for models to discover, especially at scale.

**Strengths:**

- Rigorous empirical analysis: The paper provides tight power-law fits across the tested regime, demonstrating clear architecture-dependent scaling exponents with convincing statistical evidence.
- Clear presentation of core results: The main findings about equivariant architectures exhibiting better scaling behavior than non-equivariant models are presented clearly.
- Practical insights: The finding that data and model size should scale in tandem for compute-optimal training is actionable regardless of architecture choice.
- Important research question: Investigating whether fundamental inductive biases matter at scale is highly relevant to the broader machine learning community.

**Weaknesses:**

- Severely limited training regime: Training on only a single epoch is a poor experimental choice that prevents data augmentation and fails to test the most interesting hypothesis from Brehmer 2025—that equivariance benefits may disappear with sufficient augmentation over multiple epochs. While the authors may draw inspiration from language model training, molecular datasets are orders of magnitude smaller, making this analogy weak.
- Insufficient scale: The maximum compute budget of ~30 GPU hours is quite small for claiming insights about "large-scale" behavior on molecular datasets. True scaling studies typically involve orders of magnitude more compute.
- Limited symmetry analysis: What the paper calls "unconstrained MLP" is actually translation-invariant (operating on relative positions). The study only assesses rotational equivariance on top of translational invariance, not the full contribution of geometric symmetries. This should be explicitly clarified as it affects the interpretation of results.
- Unexplained contradictions with recent work: The conclusions don't adequately address the success of models like Orb, which achieve near state-of-the-art performance on MatBench Discovery despite being only translational equivariant (like the unconstraint MPNN in this paper). This weakens the paper's strong claims about the necessity of equivariance at scale. Could it be that the non-rotationally-equivariant model chosen by the authors is a worse choice than Orb?

**Questions:**

- How do the resulting models compare to state-of-the-art models on the OpenMol dataset? That would be good to know how well the results represent real-life results on such datasets.
- Have the authors considered taking a model like eSEN and manipulating just its equivariant convolution layers to be unconstrained? That would remove some of the impact of architectural design on the comparison.

---

> ### Author Response · Authors · 2025-11-21
> **Official Comments by Authors**
>
> We thank reviewer RXkz for dedicating their time to review our work. We are pleased that they found our work to have practical insights. We would like to address the raised weaknesses and questions below.
>
> ***Weakness 1: Single epoch training and the lack of data augmentation.***
>
> Please see the updated version of our manuscript (section 4.5, figs 8, 9). In particular, we trained the models on a 1% of the full data over 100 epochs. We compared unconstrained models trained with data augmentation to eSEN. Our results show that both of them recover the same compute-optimal scaling laws as in one-pass training over the full dataset, in which eSEN has a larger scaling exponent. This shows that "equivariance benefits do NOT disappear with augmentation over multiple epochs" when one is considering scaling laws. At best, data-augmentation helps _preserve_ the same scaling behaviour for larger compute budget.
>
> ***Weakness 2: Insufficient scale***
>
> We go up to 100 GPU hour in single epoch training.
> This is on par with largest models trained in this domain; please refer to Table 1 in [4]. Also, please see our general response to question 1 and 2 above.
>
> ***Weakness 3: Limited symmetry analysis***
>
> The symmetries under consideration are permutation (through body order) and rotational symmetry (through tensor order). All of our models work with translation invariant representations of the input. We have clarified this in the revised paper; please see line 79 in the updated manuscript.
> For completeness, thanks to the reviewer's suggestion we also added a small experiment where we use raw coordinates as node inputs for the unconstrained model; see Appendix J. We see a performance drop.
>
>
> ***Weakness 4: Unexplained contractions with recent work like Orb***
>
> Our unconstrained model indeed resembles the Orb's architecture.
>
> However, Orb [2] is extensively _pre-trained_ using a denoising objective before being fine-tuned on the downstream force-field task. Therefore, it is not directly comparable to the models in our study, which do not use any form of pretraining. We note that denoising-based pretraining is becoming a common practice for both equivariant [1] and non-equivariant models [2] in force-field prediction. We have acknowledged this in the limitation section in our manuscript and plan to investigate scaling laws under this pretraining regime in future work. For clarity, we have added additional comments regarding this discrepancy in the revised version; please see lines 326–329.
>
> ***Question 1***
> >How do the resulting models compare to state-of-the-art models on the OpenMol dataset? That would be good to know how well the results represent real-life results on such datasets.
>
> eSEN, GemNet and unconstrained MPNN (similar to ORB) which we study in this work indeed achieve competitive results according to their leaderboard performance. Please see https://huggingface.co/spaces/facebook/fairchem_leaderboard and https://www.orbitalindustries.com/posts/orbmol-extending-orb-to-molecular-systems.
>
> ***Question 2***
> > Have the authors considered taking a model like eSEN and manipulating just its equivariant convolution layers to be unconstrained? That would remove some of the impact of architectural design on the comparison.<
>
> While eSEN is SO(3) equivariant, due to edge-based alignments in message updates these convolution operations are SO(2) equivariant. The implications of making these layers unconstrained is not obvious.
> We are avoiding any new model design in our study and only using representative architectures that have proven their utility through their impact. Moreover, it would be difficult to draw convincing conclusions about the scaling law of unconstrained vs equivariant networks if the architectures under study are obscure or arbitrary.
> Please also see question 5 in our general response above.
>
> ***References***
>
> [1] Yi-Lun Liao, Tess Smidt, Muhammed Shuaibi, and Abhishek Das. Generalizing denoising to non-equilibrium structures improves equivariant force fields. Transactions on Machine Learning Research, 2024a. ISSN 2835-8856. URL https://openreview.net/forum?id=whGzYUbIWA.
>
> [2] Mark Neumann, James Gin, Benjamin Rhodes, Steven Bennett, Zhiyi Li, Hitarth Choubisa, Arthur Hussey, and Jonathan Godwin. Orb: A fast, scalable neural network potential, 2024. URL https://arxiv.org/abs/2410.22570.
>
> [3] Levine, D.S., Shuaibi, M., Spotte-Smith, E.W.C., Taylor, M.G., Hasyim, M.R., Michel, K., Batatia, I., Csányi, G., Dzamba, M., Eastman, P. and Frey, N.C., 2025. The open molecules 2025 (omol25) dataset, evaluations, and models. arXiv preprint arXiv:2505.08762.
>
> [4] Duval, A.A., Schmidt, V., Hernández-Garcıa, A., Miret, S., Malliaros, F.D., Bengio, Y. and Rolnick, D., 2023, July. Faenet: Frame averaging equivariant gnn for materials modeling. In International Conference on Machine Learning (pp. 9013-9033). PMLR.

---

### Official Review · Reviewer_SZDj · 2025-10-31

**Soundness:** 2
**Presentation:** 3
**Contribution:** 2
**Rating:** 4
**Confidence:** 4

**Summary:**

The paper presents an empirical study on the scalability of equivariant models vs non-equivariant models (w.r.t.  SO(3) symmetry group) for the molecular force fields task (predicting energy and forces of molecules). More specifically, the authors show how models like GemNet, eSEN, and EGNN tend to scale better than unconstrained GNN architectures on the OpenMol dataset. They study the scalability of these models across multiple access, including: model parameters, PFLOPs, and GPU hours, against validation loss of the molecular force fields task.

**Strengths:**

* I think the problem being studied is interesting and quite relevant to the current ongoing discussion on equivariant vs non-equivariant models design, in the area of geometric deep learning.
* Interesting to see that different equivariant models have different scalability behavior, for example, eSEN has lower validation loss with more GPU hours compared to the GemNet architecture.

**Weaknesses:**

* I think the evidence is not sufficient to support the claims; some rewrites might be required to not make it a general claim or show more results on other tasks.
* The study is limited to a single-epoch training regime, and studying the GPU hours in the range of less than 100 hours is limited. It would be beneficial to see how this could be extended for longer training times, and if the same trend holds or not.
* Also, comparing recent models like eSEN to vanilla GNN might limit the claims of the paper, as GNN/ MPNN is an old baseline now. More unconstrianed architectures should be included (e.g., how this is applied to Graph Transformers).

**Questions:**

See above.

---

> ### Author Response · Authors · 2025-11-21
> **Official Comments by Authors**
>
> We thank reviewer SZDj for dedicating their time to review our work. We are pleased that they found our work is interesting. We would like to address the identified weaknesses below.
>
> ***Weakness 1: Insufficient evidence for general claims***
>
> We agree about not having any general claims beyond what we observe empirically. Our claims are limited to the specific domain of force-fields. Could you please clarify which specific claim is not sufficiently supported? We would be happy to adjust them.
>
> ***Weakness 2: Single-epoch training regime and limited training GPU hours.***
>
> Please see our general response about multi-epoch training and scale of our experiments. In summary, our new results in multi-epoch setting suggests that going to about 100x our current compute using multi-epoch training will maintain the scaling of loss vs compute for each model size.
>
> ***Weakness 3: Choice of architectures***
>
> To our knowledge, there are no non-equivariant graph transformers that perform well enough for interatomic potentials, especially when equivariant targets like forces are in the equation. In fact, our choice for vanilla GNN is close to some of current SOTA non-equivariant architectures for force fields [1, 2]. Please also see our general response on the choice of architecture.
>
> ***References***
>
> [1] Neumann, M., Gin, J., Rhodes, B., Bennett, S., Li, Z., Choubisa, H., Hussey, A. and Godwin, J., 2024. Orb: A fast, scalable neural network potential. arXiv preprint arXiv:2410.22570.
>
> [2] Duval, A.A., Schmidt, V., Hernández-Garcıa, A., Miret, S., Malliaros, F.D., Bengio, Y. and Rolnick, D., 2023, July. Faenet: Frame averaging equivariant gnn for materials modeling. In International Conference on Machine Learning (pp. 9013-9033). PMLR.

---

### Official Review · Reviewer_AQ3A · 2025-11-01

**Soundness:** 3
**Presentation:** 4
**Contribution:** 3
**Rating:** 8
**Confidence:** 3

**Summary:**

This work introduces large-scale experimental results illustrating scaling laws for validation loss across equivariant message passing architectures and for NNIP tasks. As a result, several practical observations are made, including optimal architecture choices based on available compute.

**Strengths:**

The paper tackles an important and timely problem, studying the effect of higher-order equivariant features on accuracy/compute tradeoffs. It presents large-scale experiments and reports empirical scaling behavior with clear ablations. The approaches used in the study are well motivated and clearly written.

**Weaknesses:**

Some claims appear stronger than what the experiments directly support.

**Questions:**

**Q1. Generalization.** The claim on line 80

> While our study is limited in scope to Special Euclidean symmetry of neural interatomic potentials and force fields, as well as a few representative architectures, there is no reason to believe the results should remain confined to this particular domain and the choice of efficient equivariant models.

appears at odds with the line 843 on related works stating

>Despite using the eSEN same backbone, Wood et al. (2025) report that, for dense models 5, the compute-optimal strategy scales model size N faster than data size D, whereas in our setting we observe nearly equal scaling between N and D; though the tasks are different

I suggest that the authors reconsider the claims of generalization of these scaling curves beyond the scope of the present work.

**Q2. Undefined acronyms.** Please be sure to introduce acronyms like NNIPs that may be familiar to those working on these specific tasks, but not to a more general audience.

**Q3. Energy vs. direct-force training.** Can the authors please substantiate the claim in line 138

> While it is sufficient to learn the energy for predicting conservative forces, direct force prediction is significantly more scalable.

**Q4. Transformer instability.** Can the authors provide evidence for the line 149 where they observed

> instability issues when scaling vanilla transformers for this task, we focused on message-passing architectures.

---

> ### Author Response · Authors · 2025-11-21
> **Official Comments by Authors**
>
> We thank reviewer AQ3A for their assessment of our paper. We are pleased that they found our study to be timely and important. We now address the remaining questions below.
>
> ***Question 1: Generalization***
>
> There are some practices in Wood et al. (2025) that could be the cause of this conflicting result: for example, it seems (from the code and the paper) that they do not use muP scaling for the hyper-parameters, they use multi-epoch training with Chinchilla scaling laws; and they also do not use scheduler-free optimization. They also have a large uncertainty around their estimates. While their task is also different, we think that these choices in their study could have contributed to their conclusions.
>
> To be safe with our claims, following reviewer's suggestion, we have removed the referenced sentence.
>
> ***Question 2: Undefined acronyms.***
>
> Please see line 71 in our updated paper.
>
> ***Question 3: Energy vs. direct-force training***
>
> Energy-only training can produce conservative forces through automatic differentiation, the training involves backpropagation through this process and it is much more compute intensive. An emerging practice is to train both using forces and energy so as to benefit from the dense force labels in training. In post-training one could then remove the the force prediction head and fine-tune the model to predict forces using energy [1, 2].
>
> ***Question 4: Transformer instability.***
>
> Please see appendix K in our updated manuscript. We trained a GPT-style encoder and it didn't perform any learning over the training period.
>
> ***References***
>
> [1] Wood, B.M., Dzamba, M., Fu, X., Gao, M., Shuaibi, M., Barroso-Luque, L., Abdelmaqsoud, K., Gharakhanyan, V., Kitchin, J.R., Levine, D.S. and Michel, K., 2025. UMA: A Family of Universal Models for Atoms. arXiv preprint arXiv:2506.23971.
>
> [2] Amin, I., Raja, S. and Krishnapriyan, A.S., Towards Fast, Specialized Machine Learning Force Fields: Distilling Foundation Models via Energy Hessians. In The Thirteenth International Conference on Learning Representations.

---

> > ### Comment · Reviewer_AQ3A · 2025-11-26
> > **Official Response by Reviewer AQ3A**
> >
> > I thank the authors for their response. I have also read and acknowledge the other reviewers concerns. I maintain my score and reasoning.
> >
> > I may suggest providing references either to [3-4] that provide a more detailed discussion on direct/conservative force prediction, e.g., on accuracy/scalability trade-offs.
> >
> > [3] Bigi, Filippo, Marcel F. Langer, and Michele Ceriotti. “The Dark Side of the Forces: Assessing Non-Conservative Force Models for Atomistic Machine Learning.” arXiv, 16 Dec. 2024, arxiv.org/abs/2412.11569.
> >
> > [4] Fu, Xiang, et al. “Learning Smooth and Expressive Interatomic Potentials for Physical Property Prediction.” arXiv, 17 Feb. 2025, arxiv.org/abs/2502.12147.

---

> > > ### Author Response · Authors · 2025-11-27
> > > **Response**
> > >
> > > Dear Reviewer AQ3A,
> > >
> > > We appreciate your positive rating and feedback. We agree that the accuracy/speed tradeoff of direct/conservative force prediction requires more explicit discussion. Accordingly, we have added two sentences and the recommended citations to address this point in the revised manuscript (lines 143–145).

---

### Official Review · Reviewer_58Kv · 2025-11-03

**Soundness:** 1
**Presentation:** 3
**Contribution:** 2
**Rating:** 2
**Confidence:** 5

**Summary:**

The authors study how machine learning interatomic potentials (MLIPs) scale with data, parameters, and compute, comparing architectures with increasing degrees of equivariance. They train on the OMol neutral-molecule subset, treat atoms as “tokens,” and adopt a single-epoch training regime to mirror LLM practice. They evaluate scaling against both FLOPs and GPU-hours, arguing this better reflects practical costs for (often less GPU-friendly) equivariant models. Empirically, they report clear power-law behavior and that scaling exponents grow with architectural equivariance, so performance gaps widen at larger scales; they also observe that compute-optimal training should scale model size and dataset size in tandem. A symmetry loss regularizer improves sample efficiency but does not match the benefits of fully equivariant architectures.

**Strengths:**

1. The paper applies the modern neural-scaling methodology to neural interatomic potentials, an important area in AI-driven chemistry.  This fills a gap for the field.
2. The paper is well written and easy to follow.

**Weaknesses:**

As the first paper (that I am aware of) to investigate the neural scaling law in the MLIP domain, I think the standard / stakes are high, and incorrect claims could lead to consequences in following model design. In this paper, the authors draw a lot of methodology directly from NLP, but I think these two fields have some fundamental differences. I found the following critical flaws in the three aspect of scaling law: Compute (1, 2, 3), Data (3, 4), and Parameters (5), and additional issues (6,7)
1. Insufficient training / compute to claim asymptotic scaling: The scaling-law fits are based on small training budgets.  From Figure 1 and Section 4, the largest models run correspond to only \~10^1 GPU-hours (per run) or \~10^3 PFLOPs of compute.  This is tiny compared to typical scaling-law studies, which span orders-of-magnitude more compute (e.g. billions of tokens or thousands of TPU/GPU years). Even for MLIP training, this is too small: the eSEN small direct model is trained for around 10^5 PFLOP (80 Epochs) in OMol 4M, where the authors is training 10^3 PFLOPs on OMol 34M. That is **100x less FLOPs on 8x more data**. Moreover, each model is trained for only a single epoch through 34M samples (no repeated passes).  For neural potentials, one epoch training is extremely light: standard MLIP practice often trains tens of epochs to convergence.  The plots (e.g. Figure 1) do not show curves flattening -- the losses keep decreasing.  This suggests models are not converged even at the largest scale tried.  Fitting power laws in such a sub-convergent regime is problematic: the “effective” exponent can vary dramatically in early vs late training. In short, scaling laws are **asymptotic** statements, and here the training scale is too small to robustly infer asymptotic behavior.  This undermines the claim that explicit symmetry consistently improves scaling: with more compute, any performance gap might shrink or change.
2. Single-epoch training regime. In Section 3.1, the authors intentionally use **only one epoch** (each sample seen once) to mirror LLM practice. However, this contradicts common MLIP training, where models typically see the data many times (e.g. \~**80 epochs** in the OMol 4M baselines). Single-epoch training likely means none of the models fully fit the data distributions; indeed the validation losses are still dropping at the end of training. This choice can distort scaling-law estimates. For instance, early in training, increasing model size might seem more beneficial (higher \alpha) simply because larger models learn faster per data pass, but with multi-epoch training a smaller model could catch up. By never allowing converged fits, the authors effectively inflate the “scaling advantage” of larger/equivariant models. They justify this by wanting to “avoid confounding effects”, but do not analyze how one vs multiple epochs alters the conclusions. Without at least one multi-epoch comparison, we cannot be sure the reported power-law behaviors would hold under standard MLIP training. This is a **major methodological gap**: the conclusions about scaling hinge on a very nonstandard (for this domain) training regime.
3. Global target vs. token-wise analogy. The paper frequently draws analogies to language scaling laws, but **the supervised targets here are fundamentally different**. In language models, the loss (cross-entropy) is accumulated over **each token prediction**. Here, the primary target is the global energy of a molecule (plus per-atom forces, calculated by the gradient of the energy, thus is heavily correlated with the total energy). Each system yields **only one energy scalar despite many atoms**. The authors do treat atoms as “tokens” in counting dataset size, but this is a loose analogy: the model must capture a global property that depends on all atoms jointly. This could be a much harder task than NLP cross entropy target. Thus, as we discussed in weakness 2, seeing each atom once may not be analogous with LLM training, and the model requires much more training time to converge. As a result, statements like “we follow LLM scaling methodology” may be overreaching without addressing that molecular systems are global-structure tasks.
4. Limited dataset regime (neutral split only). The OMol dataset is a high quality and diverse dataset, but all experiments use only the **neutral-molecule subset** of OMol (\~34M samples), despite the full dataset being ~100M with diverse splits. This choice (claims to be taken due to memory limits, which if that's because of the system size, the authors could always decrease the batch size, since the batch size they chose was quite large (64) ) of using the neutral split, rather than **random sample**, could bias the domain. As described in the OMol paper, **non-equivariant models, such as GemNet, performs much better than equivariant models in EF error** when trained on all splits, such as biomolecules and electrolytes. This dataset selection clearly favors the equivariant baselines, and leads to a misleading or even incorrect results. I.e. the conclusions about “symmetry matters more at scale” could be dataset-specific. The authors should justify why neutral-only results would hold in the full OMol or other datasets (none of which is provided).
5. Unfair comparison by raw parameter count. In NLP, comparing models by parameter count is largely meaningful because mainstream LLMs use a homogeneous Transformer architecture: blocks are architecturally identical, most weights live in token-wise MLPs and attention projections, and the compute per parameter and weight reuse pattern are effectively uniform across models. As a result, plotting loss vs. parameters is reasonably apples-to-apples. In contrast, **for GNN-based MLIPs the notion of “one parameter” does not carry across architectures**: (i) parameters are shared and reused across all nodes/edges; (ii) different designs incur very different work per parameter (e.g., dihedral terms in GemNet-OC, high-order tensor ops in eSEN); (iii) body order and tensor order change effective capacity and FLOP intensity; and (iv) throughput at the same N can differ drastically. This is largely why the kappa number varies between architectures. Consequently, the Fig. 5 style plots of loss vs. N conflate capacity, compute, and inductive bias, and can be misleading. A fair comparison should be iso-compute (FLOPs or GPU-hours), or at least normalized by an architecture-dependent cost.
6. Exponent reliability and fitting issues. The fitted power-law exponents (α, β, γ) are central to the paper’s claims, but their reliability is questionable. First, as noted, the data range is relatively narrow (datasets from 10%-100%, models \~10^6-10^7 parameters): typically one requires several orders of magnitude variation to robustly determine an exponent.  Here both N and D spans are \~1-2 orders of magnitude at best. Second, the assumption $L_\inf \approx 0$ (irreducible loss) is ad hoc and may bias exponents high.  In real molecular data there should be some finite noise/error floor (analogous to the entropy in NLP); assuming zero means the model is “expected” to achieve perfect prediction eventually.  Small positive L_\inf can drastically change a fitted β or γ in a power law fit (as studied in Hoffmann et al. 2022).  Third, Figure captions reveal instability: GemNet-OC needed smoothed loss, and the authors exclude the first 1–10% of training from fits.  These steps suggest the raw curves were not clean power laws. With such variance, the numeric exponents conflict across FLOPs vs wall-time (Fig.1).  Without error bars or fit-statistics on α,β (only γ had CIs in Table 1), it’s hard to trust these values.  In summary, the power-law behavior is claimed too strongly: given the limited and noisy regime, the reported exponents may not reflect true asymptotic trends. The results could be artifacts of the experimental choices (one-shot training, hyperparams, smoothing).
7. Additional concerns: (i) The study considers only four architectures; it is not clear if these are representative. For example, EGNN and GemNet are relatively simple and old models -- how would a more modern one (PaiNN, DPA-1/2, EScAIP, even NequIP) behave? (ii) The paper also mixes different body-order and tensor-order notions without clarity. (iii) Hyperparameters (depth, widths) are tuned only at ~1M params and then scaled; it’s possible that later models were not fully optimized. (iv) The authors do not report if they repeated experiments to measure variability (confidence intervals are absent for N,D scaling). (v) Finally, the broad claim that one should “not leave symmetry to be discovered by scaling” may overstate the results. The bitter-lesson cited in the intro suggests that even biased models can eventually be outperformed by larger unconstrained ones; these experiments do not go far enough to test that (no extremely large unconstrained model was trained). In its current form, the paper’s strong conclusion about avoiding bitter lesson is not fully justified by the limited empirical data.

**Questions:**

1. Why restrict training to a single epoch?  Have you tried multi-epoch training (e.g. 10 or 80 epochs) on a smaller scale to see if exponents change?  How do you justify that one pass through data captures the scaling behavior of fully trained models?
2. How do you expect inclusion of the charged/large molecules (omitted in the neutral split) to affect your conclusions?  Could the symmetry advantages reverse or diminish in chemically diverse subsets?
3. How sensitive are your results to the learning-rate and batch-size choices? You tuned these at 1M parameters and then scaled. Is it possible that some models (e.g. the largest eSEN) were suboptimally trained?
4. The introduction cites Sutton’s “bitter lesson” about scale overtaking bias. Given your modest scaling regime, how confident are you that “we should not leave symmetry to be discovered by the model”? Could larger-scale experiments eventually reverse the trend you observe?
5. Nit: where is Figure 6?

---

> ### Author Response · Authors · 2025-11-21
> **Official Comments by Authors**
>
> We thank reviewer 58Kv for their detailed and critical feedback, which helps improve the paper. Below, we address the identified weaknesses and questions.
>
> ***Weakness 1 Insufficient training / compute to claim asymptotic scaling***
>
> New results: increasing the FLOPS with more epochs maintains the loss-vs-compute linear trend in log-log, as if we have new data. We hope this addresses the concern with training budget.
>
> Then why not study scaling law with multi-epoch (considering much larger epochs)? because as our results show when we increase compute we should invest it in both dataset and model size, but we cannot increase the model size beyond what we currently have due to memory issues. So overall the cleanest path is to consider the single epoch setting, noting that the trend holds for tens of epochs.
>
> ***Weakness 2: Single-epoch training regime.***
>
> We include multi-epoch results in the updated paper. Please see the general response and Section 4.5.
>
> ***Weakness 3: Global target vs. token-wise analogy.***
>
> There seems to be a misunderstanding: we are not primarily using global graph-level target which is the energy but **node-level targets that are atomic forces**. The training signal in force-prediction is much more dense and resembles token-level loss in next-token prediction. The energy prediction is included following the common practice, as it can facilitate a post-training procedure in which the force prediction heads are removed and conservative forces are then predicted using energy [1,2].
>
> ***Weakness 4: Limited dataset regime (neutral split only).***
>
> We have added new results on 4M split of OMol to address your concern about diversity of the selected subset.  Please also note that the neutral dataset "is intended to measure the performance
> of models on datasets the community is familiar with" according to curators of OMol. So we believe our orignal choice was sound. Nevertheless, new results do not change the story. On the technical side the issue in using the entirety of the OMol dataset is with the system/main memory and IO issues not the GPU memory. The large dataset of 100M data points is ~500GB in main memory, and one needs to stream the dataset from the hard drive rather than load it into the main memory. Many GPU clusters, including our academic cluster, restrict IO with the data-loader. With the neutral split we could load the entire dataset into memory and avoid these issues. Note that, with the exception of [1], even the neutral split is the largest dataset used to study scaling laws in geometric deep learning, including prior and concurrent works [8, 9].
>
> ***Weakness 5: Unfair comparison by raw parameter count.***
>
> We are confused about the issue being raised. We agree with the individual points, to the extent we understand them, but do not understand the conclusion about some results being misleading. Could you please elaborate? Could you also explain what you mean by "throughput" in "throughput at the same N can differ drastically"?
>
> ***Weakness 6: Exponent reliability and fitting issues.***
>
> There are several independent issues raise here that we address separately:
> - In this setting because the ground truth is produced by DFT and it is deterministic, we know that the irreducible error is zero (see also[1,7]).
> - Please note that we DO have all the confidence intervals reported in the appendix.
> - We follow the findings in [5,6] and their recommended procedure in droping the early checkpoints for a reliable scaling law. This is because "Scaling Exponents are Task-Dependent at _Late_ Training Time" [6]. That is why the smaller dataset sizes are dropped in our plots, and as we discussed above the larger dataset sizes or multi-epoch training would require simultaneous scaling of model size for optimal compute. Nevertheless our compute optimal scaling results span almost five orders of magnitude. We hope the multi-epoch training results and the experiments using even smaller models (running at the moment), address the reviewer's concern.

---

> ### Author Response · Authors · 2025-11-21
> **Official Comments by Authors**
>
> ***Weakness 7: Additional concerns.***
>
> We would like to emphasize each concern (line-by-line) as follows:
>
> i. As stated in Section 2 (line 148 to 175), our work aims to cover a wide range of design principles for equivariance modeling. This categorization is based on previous works studying expressiveness of geometric graph neural networks, and we select the most representative in each category. Our inability to scale vanilla transformer for this task rules out transformer-based models.
>
> We respectfully disagree that Gemnet is an old model. In fact, we chose GemNet-OC because the model had strong performance across molecular benchmarks (please see https://huggingface.co/spaces/facebook/fairchem_leaderboard). In particular, it outperforms other models in the same cateorgy (rotation order 1, higher body order) [3],such as PaiNN, suggested by the reviewer. Similarly, eSEN is currently the SOTA model, and comppares favorably against the suggested NequIP and other  architectures in this cateogory [13].
>
> ii. We clearly distinguish these orders in our presentations and results, e.g., table 1 in Appendix C; could you please elaborate?
>
> iii. We use muP scaling [4] of hyper-parameters, which necessitates this setup, and ensures optimality as we scale.
>
> iv. We did report confidence intervals for N,D scaling in the appendix. Please see section D and Table 7.
>
> v. Our results are limited to a specific application domain, but they do consider the largest models we can fit in a single A100 GPU memory.
> In fact, the number of parameters in our unconstrained model is 20 times higher than equivariant models, and the trends we report are clear.
>
> ***Question 1***
> > Why restrict training to a single epoch? Have you tried multi-epoch training (e.g. 10 or 80 epochs) on a smaller scale to see if exponents change? How do you justify that one pass through data captures the scaling behavior of fully trained models?<
>
> Please see the new results on multi-epoch training. The exponents remain the same for tens of epochs and then then change slowly. This suggests that multi-epoch training for 10s of epochs remains similar to single epoch training with a larger dataset, and our conclusion remains valid. A concurrent work under review [3] characterizes this relationship between multi-epoch training and larger dataset to a larger extent.
>
> ***Question 2***
> > How do you expect inclusion of the charged/large molecules (omitted in the neutral split) to affect your conclusions? Could the symmetry advantages reverse or diminish in chemically diverse subsets? <
>
> Please see Appendix G in our updated manuscript. In general, the advantages of symmetry-aware design are preserved even when the training data includes charged and/or large molecular species.
>
> ***Question 3***
> > How sensitive are your results to the learning-rate and batch-size choices? You tuned these at 1M parameters and then scaled. Is it possible that some models (e.g. the largest eSEN) were suboptimally trained? <
>
> Using muP scaling it is common to assume learning rates remain optimal across scales [4]. With batch-size we observe that smaller batch sizes produce better results (see figure 10). However, this significantly increases the run-time, and so we fix the batch size across architectures and scales.
>
> ***Question 4***
> > The introduction cites Sutton’s “bitter lesson” about scale overtaking bias. Given your modest scaling regime, how confident are you that “we should not leave symmetry to be discovered by the model”? Could larger-scale experiments eventually reverse the trend you observe?
>
> Our models are comparable to large foundation models that have been trained for this domain. Please see the leaderboard of models at [11] and https://matbench-discovery.materialsproject.org/. In the tables, the parameter counts of all listed models do not exceed 100M. Moreover, Orb-v3 is a non-equivariant message-passing network with only 25M parameters, whereas our largest unconstrained model has up to 80M parameters; note that Orb-v3 was pre-trained with a denoising objective.
>
> We are indeed aware of the weight-sharing property of equivariant models. Accordingly, in our experiments, unconstrained models are significantly larger than their equivariant counterparts, with the largest gap reaching nearly a factor of 20. Therefore, we ensure a fair comparison in this setting.
>
> Altogether, we believe the claims are based on clear evidence, although limited to a single task.
>
> ***Question 5***
> > Nit: where is Figure 6?
>
> We're sorry for this! There was a typo error in our LaTeX source. Specifically, we accidentally used \captionof instead of \caption in Figure 6 (which appeared as Figure 7 in the original submission), causing the subsequent figure numbering to shift. This has been corrected in the revised manuscript.

---

> ### Author Response · Authors · 2025-11-21
> **Official Comments by Authors**
>
> ***References***
>
> [1] Wood, B.M., Dzamba, M., Fu, X., Gao, M., Shuaibi, M., Barroso-Luque, L., Abdelmaqsoud, K., Gharakhanyan, V., Kitchin, J.R., Levine, D.S. and Michel, K., 2025. UMA: A Family of Universal Models for Atoms. arXiv preprint arXiv:2506.23971.
>
> [2] Amin, I., Raja, S. and Krishnapriyan, A.S., Towards Fast, Specialized Machine Learning Force Fields: Distilling Foundation Models via Energy Hessians. In The Thirteenth International Conference on Learning Representations.
>
> [3] Anonymous. Larger datasets can be repeated more: A theoretical analysis of multi-epoch scaling in linear regression. In Submitted to The Fourteenth International Conference on Learning Representations, 2025. URL https://openreview.net/forum?id=0CXjpAxHUE. under
> review.
>
> [4] Ge Yang, Edward Hu, Igor Babuschkin, Szymon Sidor, Xiaodong Liu, David Farhi,
> Nick Ryder, Jakub Pachocki, Weizhu Chen, and Jianfeng Gao. Tuning large neu-
> ral networks via zero-shot hyperparameter transfer. In M. Ranzato, A. Beygelzimer,
> Y. Dauphin, P.S. Liang, and J. Wortman Vaughan (eds.), Advances in Neural In-
> formation Processing Systems, volume 34, pp. 17084–17097. Curran Associates, Inc.,
> 2021. URL https://proceedings.neurips.cc/paper_files/paper/2021/
> file/8df7c2e3c3c3be098ef7b382bd2c37ba-Paper.pdf
>
> [5] Leshem Choshen, Yang Zhang, and Jacob Andreas. A hitchhiker’s guide to scaling law estimation. In International Conference on Machine Learning, 2025.
>
> [6] Bordelon, B., Atanasov, A. and Pehlevan, C., 2024, July. A Dynamical Model of Neural Scaling Laws. In International Conference on Machine Learning (pp. 4345-4382). PMLR.
>
> [7] Brehmer, J., Behrends, S., De Haan, P. and Cohen, T., 2024. Does equivariance matter at scale?. arXiv preprint arXiv:2410.23179
>
> [8] Frey, N.C., Soklaski, R., Axelrod, S., Samsi, S., Gomez-Bombarelli, R., Coley, C.W. and Gadepally, V., 2023. Neural scaling of deep chemical models. Nature Machine Intelligence, 5(11), pp.1297-1305.
>
> [9] Trikha, A., Chu, K., Gosai, A., Szachta, P. and Weiner, E., 2025. Scaling Laws for Neural Material Models. arXiv preprint arXiv:2509.21811.
>
> [10] Johannes Klicpera, Florian Becker, and Stephan G¨unnemann. Gemnet: Universal directional graph neural networks for molecules. In Conference on Neural Information Processing (NeurIPS), 2021
>
> [11] Levine, D.S., Shuaibi, M., Spotte-Smith, E.W.C., Taylor, M.G., Hasyim, M.R., Michel, K., Batatia, I., Csányi, G., Dzamba, M., Eastman, P. and Frey, N.C., The Open Molecules 2025 (OMol25) Dataset, Evaluations, and Models. 2025. Preprint at https://arxiv.org/abs/2505.08762.
>
> [12] Gasteiger, J., Shuaibi, M., Sriram, A., Günnemann, S., Ulissi, Z., Zitnick, C.L. and Das, A., 2022. GemNet-OC: developing graph neural networks for large and diverse molecular simulation datasets. arXiv preprint arXiv:2204.02782.
>
> [13] Fu, X., Wood, B.M., Barroso-Luque, L., Levine, D.S., Gao, M., Dzamba, M. and Zitnick, C.L., 2025. Learning smooth and expressive interatomic potentials for physical property prediction. arXiv preprint arXiv:2502.12147.

---

### Author Response · Authors · 2025-11-21
**General Response (Part 2)**

**4. Why neutral subset of OMol? Are there any implications?**

We used neutral split because it reflects the performance of widely used models in this domain. Particularly, here we quote the claim of [9]: **"This split is intended to measure the performance
of models on datasets the community is familiar with, without worrying about the complexity of charge and spin"**

Nevertheless, in the updated version of the paper, we have included results on a smaller subset that includes charged molecules; the scaling behaviour remains similar; see Appendix G.

**5. Are your choice of architectures representing the best equivariant and unconstrained models? How/why did you select these specific models?**

Yes, our choices represent performant models in different categories. This choice is supported with performance on leaderboards, popularity and coverage of different categories. The categories are based on previous works studying expressiveness of geometric graph neural networks, and we select the most representative in each category, as discussed in the paper. eSEN is at the top of leaderboard here [4], and our unconstrained model is similar to ORB [5]. Our decision to focus on message passing methods rather than transformer models was due to our inability to scale vanilla transformer (see new reported results for scaling of vanilla transformer in Appendix K).

### References
[1] Muennighoff, N., Rush, A., Barak, B., Le Scao, T., Tazi, N., Piktus, A., Pyysalo, S., Wolf, T. and Raffel, C.A., 2023. Scaling data-constrained language models. Advances in Neural Information Processing Systems, 36, pp.50358-50376.

[2] Lin, L., Wu, J. and Bartlett, P.L., 2025. Improved Scaling Laws in Linear Regression via Data Reuse. arXiv preprint arXiv:2506.08415

[3] Anonymous. Larger datasets can be repeated more: A theoretical analysis of multi-epoch scaling in linear regression. In Submitted to The Fourteenth International Conference on Learning Representations, 2025. URL https://openreview.net/forum?id=0CXjpAxHUE. under
review.

[4] https://huggingface.co/spaces/facebook/fairchem_leaderboard

[5] https://www.orbitalindustries.com/posts/orbmol-extending-orb-to-molecular-systems

[6] Wood, B.M., Dzamba, M., Fu, X., Gao, M., Shuaibi, M., Barroso-Luque, L., Abdelmaqsoud, K., Gharakhanyan, V., Kitchin, J.R., Levine, D.S. and Michel, K., 2025. UMA: A Family of Universal Models for Atoms. arXiv preprint arXiv:2506.23971.

[7] Batatia, I., Benner, P., Chiang, Y., Elena, A.M., Kovács, D.P., Riebesell, J., Advincula, X.R., Asta, M., Avaylon, M., Baldwin, W.J. and Berger, F., 2025. A foundation model for atomistic materials chemistry. The Journal of Chemical Physics, 163(18).

[8] Kim, K., Kotha, S., Liang, P. and Hashimoto, T., 2025. Pre-training under infinite compute. arXiv preprint arXiv:2509.14786.

[9] Levine, D.S., Shuaibi, M., Spotte-Smith, E.W.C., Taylor, M.G., Hasyim, M.R., Michel, K., Batatia, I., Csányi, G., Dzamba, M., Eastman, P. and Frey, N.C., 2025. The open molecules 2025 (omol25) dataset, evaluations, and models. arXiv preprint arXiv:2505.08762

[10] Hestness, J., Narang, S., Ardalani, N., Diamos, G., Jun, H., Kianinejad, H., Patwary, M.M.A., Yang, Y. and Zhou, Y., 2017. Deep learning scaling is predictable, empirically. arXiv preprint arXiv:1712.00409.

[11] Frey, N.C., Soklaski, R., Axelrod, S., Samsi, S., Gomez-Bombarelli, R., Coley, C.W. and Gadepally, V., 2023. Neural scaling of deep chemical models. Nature Machine Intelligence, 5(11), pp.1297-1305.

[12] Trikha, A., Chu, K., Gosai, A., Szachta, P. and Weiner, E., 2025. Scaling Laws for Neural Material Models. arXiv preprint arXiv:2509.21811.

---

### Author Response · Authors · 2025-11-21
**General Response (Part 1)**

We thank all reviewers for their valuable feedback which has helped improved the paper. Below we briefly answer key concerns and related new results and changes to address them.

The new results are presented within the updated manuscript. Please consult the blue-highlighted sections for the location of these additions.

**1. Do our conclusions hold in multi-epoch setting?**

Short answer is yes, they remain correct for up to tens of epochs, which is the current norm in some related foundation models; e.g., see [6, 7].
We present new results showing that the linear trend of loss vs log-compute remains intact if instead of using the entire dataset only use 1% of dataset for 100 epochs (see _Section 4.5_ in the updated paper). This agrees with [1, 2, 3] which shows that when the number of training epochs is relatively small, the order of the loss remains consistent with that of one-pass SGD under the same number of iterations. For unconstrained models, the linear trend holds for fewer epochs due to overfitting compared to equivariant models. Using a small subset helps highlight the effect of overfitting in multi-epoch setting.

**2. What about data-augmentation? Can it close the gap in scaling laws?**

Since the data is not canonicalized, data augmentation only makes sense in multi-epoch training.
Our new results show that data-augmentation helps the unconstrained model retain its linear trend similar to equivariant networks. However, the power-law itself does not change.
This basically means that as compute increases, the performance gap between unconstrained model with data augmentation and equivariant architectures continues to grow, and our conclusions remain valid; see _Section 4.5_ in the updated paper.

In addition to training time augmentation, we also added new results on _test-time_ augmentation for the unconstrained model; see Appendix H. The results suggest that test-time augmentation can help by improving the multiplicative coef. rather than the exponent in the scaling law of unconstrained model. Moreover, we see that the benefit of test-time augmentation saturates quickly for this task.


**3. Is our model, dataset and compute large enough to support our conclusions? Do we have enough variation in orders of magnitude for each dimension?**

Our compute optimal scaling law spans almost five orders of magnitude. To further extend this on either end we need to consider increase/decrease in "both" the model size and the dataset size, otherwise we cannot claim/identify compute-optimal scaling. Even if we use multi-epoch training to virtually increase the dataset size, our largest models already saturate the 40GB GPU memory of A100 GPUs and we cannot scale both. We are running experiments using _smaller_ model sizes (down to single channel). We cannot further decrease the dataset size because one needs enough training to move beyond small-data regions where models can only perform as well as random guessing; see [10]. So we have exhausted our options in scaling for producing our main compute-optimal frontier results.
Separate results for scaling compute with data _alone_ using multi-epoch is possible as discussed earlier and we have included those. Also note that the scale of data used in this study is substantially larger than that of prior or concurrent works in this area [11, 12] ; we have close to one Billion tokens, which  exceeds the token counts used in some small- to medium-scale studies in NLP [8].

---

### Meta-Review · Area_Chair_JXQA · 2026-01-05

**Summary:**

This paper presents an empirical study of scaling laws for baseline models versus equivariant counterparts in a specific application: learning interatomic potentials. They conclude that the equivariant models scale better with the model size than the non-equivariant counterparts.

This paper had original scores of (2, 4, 4, 8). Since the paper is entirely empirical, most of the criticisms focused on whether the experiments give sufficient evidence to support the paper's conclusions. The authors responded by doing more computational experiments, and explaining that the experiments they provided are on-par with state-of-the-art experiments in the learning interatomic potentials literature.

I think the author's response is very comprehensive and I am favorable to accept the paper. However, since the original scores are so low, and the review with the most negative score is the most comprehensive and confident, I would like another AC to take a second look.

**Reviewer Concerns:**

Most of the reviewers' concerns had to do with the scope of the experimental evaluation. The main criticism was that the original experiments were only trained in one epoch. The authors addressed that point by reducing the size of the data and redoing the experiments for more epochs, obtaining similar results. I think that's a fair compromise.

Most of the other concerns are pretty well summarized in the author's global response. However, I think it'd be hard to know what concerns are still outstanding without asking the reviewers directly.

**Reviewer Scores:**

I think probably reviewer 58Kv would have raised their score, since the authors responded point by point to their criticisms.

Maybe reviewer RXkz would have raised their score, depending on whether they thought that the authors addressed the claims by Brehmer 2025 sufficiently well.

---

### Decision · Program_Chairs · 2026-01-26

Accept (Poster)